# Variability of surface gravity wave field over a realistic cyclonic eddy

Gwendal Marechal[1] and Charly de Marez[2]

[1]Univ. Brest, CNRS, Ifremer, IRD, Laboratoire d'Océanographie Physique et Spatiale, Brest, France
[2]California Institute of Technology, Pasadena, California

**Correspondence:** gwendal.marechal@ifremer.fr

**Abstract.** Recent remote sensing measurements and numerical studies have shown that surface gravity waves interact strongly with small-scale open ocean currents. Through these interactions, the significant wave height, the wave frequency, and the wave direction are modified. In the present paper, we investigate the interactions of surface gravity waves with a large and isolated realistic cyclonic eddy. This eddy is subject to instabilities leading to the generation of specific features both at the mesoscale and submesoscale ranges. We use the WAVEWATCH III numerical framework to force surface gravity waves in the eddy before and after its destabilization. In the wave simulations the source terms are deactivated and waves are initialized with different wave intrinsic frequencies. The study of these simulations illustrates how waves respond to the numerous kinds of instabilities in the large cyclonic eddy from a few hundred to a few ten of kilometers. Our findings show that the spatial variability of the wave direction, the mean period, and the significant wave height is very sensitive to the presence of submesoscale structures resulting from the eddy destabilization. The intrinsic frequency of the incident waves is key in the change of the wave direction resulting from the current-induced refraction and in the location, from the boundary where waves are generated, of the maximum values of significant wave height. However, for a given current forcing, the maximum values of the significant wave height are similar regardless of the frequency of the incident waves. In this idealized study it has been shown that the spatial gradient of waves parameters are sharper for simulations forced with the destabilized eddy. Because the signature of currents on waves encodes important information of currents, our findings suggest that the vertical vorticity of the current could be statistically estimated from the significant wave height gradients down to very fine spatial scale. Furthermore, this paper shows the necessity to include currents in parametric models of sea state bias; using a coarse-resolution eddy field may severely underestimate the sea-state-induced noise in radar altimeter measurements.

## 1  Introduction

The ubiquity of mesoscale (10-100 km) and submesoscale (1-10 km) eddies, fronts, and filaments in the surface of the ocean, leads to a strong variability in the wave field generated by wind (waves): waves-current interactions result in a change of significant wave height ($H_s$), frequency, and direction (Phillips (1977) and Mei (1989)).

From these modulations, it has been proved recently, thanks both to field measurements and numerical simulations, that the effects of currents on waves induce strong regional inhomogeneity of the wave field (Romero et al., 2017, 2020). In particular, Ardhuin et al. (2017) showed using realistic numerical simulations that the spatial variability of $H_s$ is closely linked to surface Kinetic Energy (KE) at the mesoscale range. Quilfen et al. (2018); Quilfen and Chapron (2019); Marechal and Ardhuin (2021)

used high resolution $H_s$ measurements from altimetry and highlight the close link between the surface current gradients and the significant wave height gradients ($\nabla H_s$). Villas Bôas and Young (2020) showed numerically, in the absence of wave dissipation and wind momentum input, that the current-induced refraction is necessarily induced by the solenoidal component of the surface currents (vorticity). Finally, Villas Bôas et al. (2020), under the same assumptions, emphasized a relationship between the vertical vorticity of the flow and the $\nabla H_s$.

Beyond a simple modification of the wave kinematics, the surface currents seem to increase the deep-water breaking wave probability and the related air–sea fluxes (Romero et al., 2017, 2020). The reader can refer to the instantaneous numerical outputs of Romero et al. (2020) and notify the local effect of the sharp current gradients on the simulated whitecap coverage (see Fig. 5d and Fig. 5i of the same reference). Wave breaking at the air-sea interface is the major source of momentum and heat exchanges between the atmosphere and the ocean (Cavaleri et al., 2012) or gas and sea spray production (Monahan et al., 1986; Bruch et al., 2021). Therefore, surface mesoscale and submesoscale currents have a significant impact on air-sea fluxes (momentum, gas, heat, sea-spray, ...) through their interactions with the wave field.

In the ocean and particularly in western boundary currents, eddies are ubiquitous from the mesoscale to the submesoscale range (Chelton et al., 2007, 2011; Gula et al., 2015b; McWilliams, 2016; Rocha et al., 2016). The interactions between the eddy field and the waves are thus of primary importance for the global distribution of wave properties. In the present study, we analyze numerically the effects of an isolated realistic eddy on the wave properties ($H_s$, mean period, and direction). Former similar works have been already performed, but only for idealized eddy cases (Gaussian profiles, see Mapp et al. (1985); Mathiesen (1987); White and Fornberg (1998); Holthuijsen and Tolman (1991); Gallet and Young (2014)), with a particular attention for the evolution of the wave direction. However, the structure of eddies in the ocean can strongly differ from textbook analytical idealized profiles (Le Vu et al., 2018; de Marez et al., 2019), making the study of the interactions between the waves and eddy with a Gaussian shape an unrealistic framework. Indeed, the instabilities occurring in a large and isolated eddy result in the strong production of energy in the oceanic submesoscales range (Hua et al., 2013; de Marez et al., 2020b) which would interact strongly with waves. Furthermore, most of the previous studies solely focused on the refraction induced by an eddy without discussing on the modulation of other wave parameters ($H_s$ or mean wave period, Mapp et al. (1985); White and Fornberg (1998); Gallet and Young (2014)). Here, our goal is to investigate the long-term mean effects of an isolated cyclonic eddy with a realistic shape (highly dynamical at the meso- and submesoscale range) on the wave properties. We demonstrate that wave field characteristics are strongly modified by the presence of the eddy and that those changes are even more significant for an eddy field with dynamics in the meso- and the submesoscale range.

This study highlight the importance to work with vortex fields with realistic spatial structures rather than with idealized eddy with a Gaussian shape. For example, in a real ocean, the resulting deviation of the waves from the great circle path due to eddy-induced refraction is certainly underestimated when eddies are considered as Gaussian (Gallet and Young, 2014). Also, previous studies in eddy rings in the vicinity of western boundary current, as in the Gulf-Stream, highlighted spatial wave height gradients at the regional scale (Holthuijsen and Tolman, 1991). These spatial gradients would have been strongly underestimated due to the too coarse aspect of the vortex geometry (Gaussian shape). Also, the estimated ocean circulation from altimeter measurements are affected by noise correlated to the $H_s$ (called sea state bias). Some proposed methods to

remove the contribution of waves in altimeter measurements assume that the wave field is smooth at scales less than 90 km (Sandwell and Smith, 2005). However, the variability of $H_s$ over a realistic eddy field pattern (more realistic than a Gaussian eddy), reveals very sharp wave parameter gradients at the scale of hundred kilometers. In a current field, the assumption that

the wave field is homogeneous at the mesoscale range is therefore not appropriate. Finally, the signature of currents on waves encodes important information that could be used to infer properties of the underlying current (Huang et al. (1972); Sheres et al. (1985) or more recently, Villas Bôas et al. (2020)). The last reference showed that sharp $\nabla H_s$ can be inverted to infer the vertical vorticity ($\zeta$) field that has generated them. In the similar numerical framework of Villas Bôas et al. (2020), we will show that the statistic of the $\zeta$ field can be estimated by inverting the variability of the wave field induced by the eddy

field. Reconstructing the $\zeta$ field would be relevant for a wide range of applications such as search and rescue, plastic debris monitoring, biological activities, or short-term wave forecast among other.

The manuscript is organized as follows: in the section 2, we introduce the eddy structure based on the works of de Marez et al. (2020b), and the numerical framework WAVEWATCH III (The WAVEWATCH III ® Development Group, 2016) without source terms. Results are presented and discussed in the section 3. The limits and the perspectives of this present work close

the paper in section 4.

## 2 Method

### 2.1 A cyclonic eddy from in-situ measurements

To study the wave propagation through an eddy field, we used the current field from the simulation performed by de Marez et al. (2020b). In this study, authors performed idealized simulations, using the Coastal and Regional Ocean COmmunity model,

CROCO (Shchepetkin and McWilliams, 2005), that solves the hydrostatic primitive equations for the velocity $\mathbf{u} = (u, v, w)$, temperature $T$, and salinity $S$, using a full equation of state for seawater (Shchepetkin and McWilliams, 2011). The spatial resolutions are chosen to accurately resolve both the frontal dynamics and the forward energy cascade at the surface. The simulation is initialized with a composite cyclonic eddy as revealed by Argo floats in the northern Arabian Sea (details of the composite extraction are fully described in de Marez et al. (2019)). The eddy is intensified at the surface, but has a deep-

reaching influence down to about 1000 m depth. Its initial horizontal shape corresponds to a shielded vorticity monopole: a positive core of vorticity and a shield of negative vorticity (Fig. 1(c)). Its radius, $R = 100$ km, is large compared to the mean regional Rossby radius $R_D$ (47 km, see Chelton et al. (1998)). It is a mesoscale eddy. In the following, mentions to "submesoscale" refers to features and processes occurring at scales that are small compared to Rossby deformation radius (*i.e.* $Bu > 1$ with $Bu = \frac{R_D^2}{L^2}$).

de Marez et al. (2020b) observed that the eddy is unstable with respect to a mixed barotropic/baroclinic instability. The latter deforms the eddy, which eventually evolves into a tripole after about 4 months of simulation. Sharp fronts are subsequently generated in the surface mixed layer at the edge of the tripole. These fronts then become unstable, and this generates sub-mesoscale cyclones and filaments. Near these fronts, diapycnal mixing occurs, causing the potential vorticity to change sign locally, and symmetric instability to develop in the core of the cyclonic eddy. Despite the instabilities, the eddy is not destroyed

and remains a large-scale coherent structure for one year of simulation. A full description of instability processes can be found in de Marez et al. (2020b). Snapshots of the current velocity and vorticity of the fully developed eddy field after 210 days of simulation are represented in Fig. 1b and Fig. 1d respectively. The main core of the cyclone is surrounded by filaments, sub-mesoscale eddies and fronts, that lead to sharp vorticity gradients. This vorticity field is far from the idealised representation of eddies often considered in the literature, and is closer to reality (see *e.g.* Fig. 1 in Lévy et al. (2018) for an example of a realistic turbulent field above mesoscale eddies).

For the purpose of the present study, we consider the surface velocity fields (the simulated level closest to the ocean surface) from the simulation outputs described above. We use the initial state that represents the eddy before instabilities occur (Fig. 1a), and the state after 210 days of simulation, in which submesoscale features have been generated by the spontaneous instability of the eddy (Fig. 1b). At 210 days all instabilities have occurred (mixed barotropic/baroclinic instabilities). After 210 days, the eddy field starts to dissipate making some small-scales features disappear (de Marez et al., 2020b). Note that the use of strictly 2D surface current is an approximation of what happen in the nature. In reality, waves feel the effects of an "average current", *i.e.*, averaged over the top few metres of the water column. The maximum depth of the current where waves can interact depends on the wavelength of the waves (Kirby and Chen, 1989).

## 2.2   The wave model

To describe the dynamics of waves over the eddy described above, we use the WAVEWATCH III numerical framework (The WAVEWATCH III $^{®}$ Development Group, 2016) forced both with the initial state (Fig. 1a,c) and the fully developed state of the eddy (Fig. 1b,d). The model integrates the wave action equation

$$\partial_t N(\sigma,\theta) + \nabla.(\dot{x}N(\sigma,\theta)) + \partial_k(\dot{k}N(\sigma,\theta)) + \partial_\theta(\dot{\theta}N(\sigma,\theta)) = S, \tag{1}$$

where $N(\sigma,\theta)$ is the wave action spectrum ($N(\sigma,\theta) = \frac{E(\sigma,\theta)}{\sigma}$, with $E(\sigma,\theta)$ the two-dimensional wave energy spectrum), $\theta$ is the direction of wave propagation, $\sigma$ the wave intrinsic frequency equals to $\sqrt{gk}$ in deep water (where water depth is largely greater than wave wavelength, here $k$ is the wavenumber and is a scalar) and g is the gravitational acceleration. $\dot{x}$ is the wave action advection velocity (equal to the sum of the wave group velocity and the surface current velocity), $\dot{k}$ and $\dot{\theta}$ are the wave advection velocities in the spectral space. The expressions of $\dot{k}$ and $\dot{\theta}$ are developed from wave ray equations (Eq. (3)) and are fully given in (Phillips, 1977; Benetazzo et al., 2013; Ardhuin et al., 2017). The right hand side of Eq. (1) is the sum of the source terms describing the wind energy input, the dissipation due to the wave breaking and bottom friction, and the nonlinear energy exchange between waves.

The dispersion of the waves is described by the dispersion relationship that links $\sigma$ and $k$. In a current field, it is necessary to consider a moving frame of reference. The waves dispersion relationship is thus impacted because the current induces a Doppler shift on the wave frequency (Eq. (2)),

$$\omega = \sigma + \mathbf{k}.\mathbf{u}. \tag{2}$$

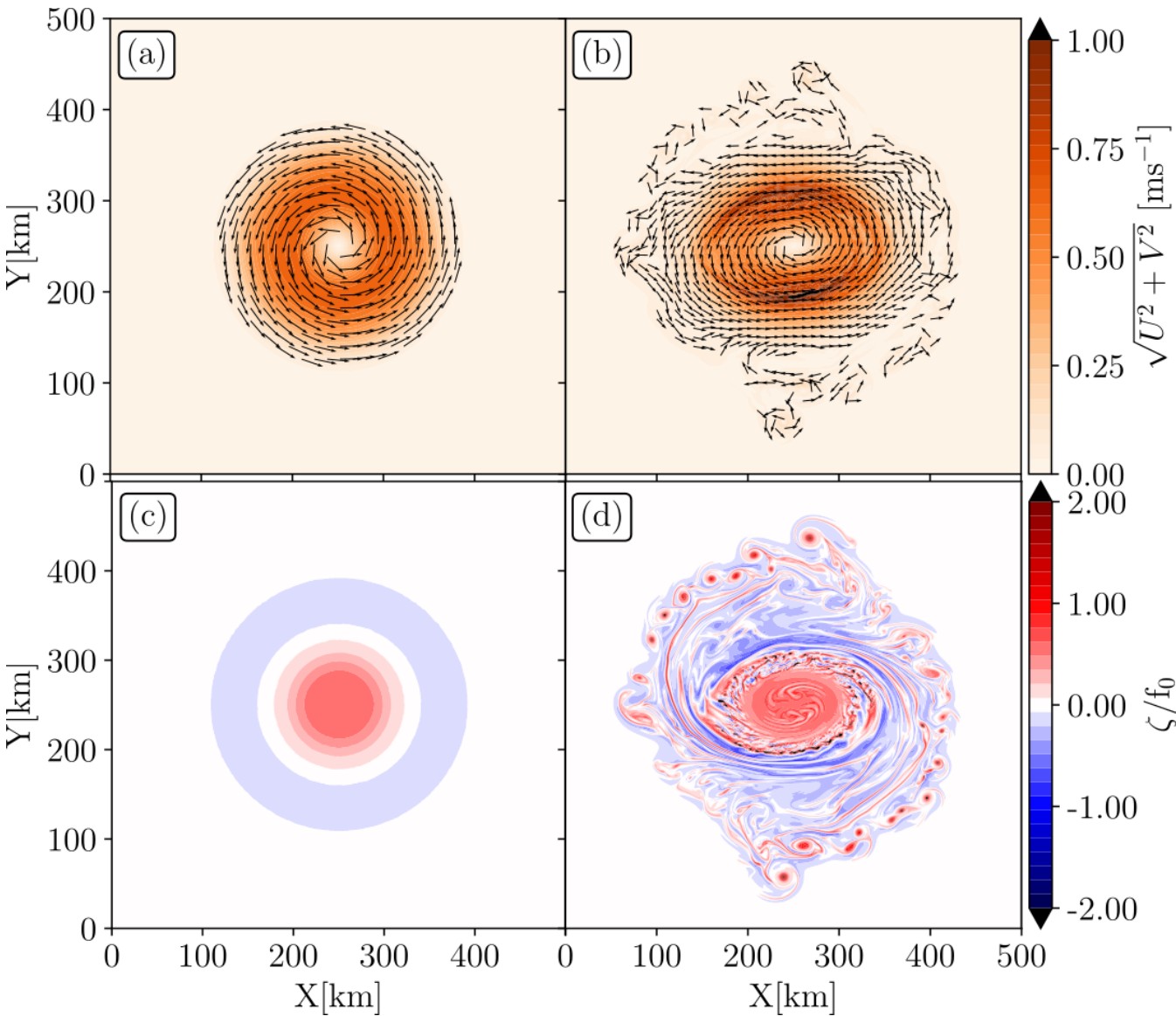

**Figure 1.** Surface current intensity and direction for the initial/Gaussian eddy (panel a) and after 210 days of destabilization (panel b). Their associated normalized relative vorticity ($\zeta=\partial_x V - \partial_y U$) are given in panel (c) and (d). The Coriolis parameter is kept constant in the simulations: $f_0 = 5.2 \times 10^{-5}\,\mathrm{s}^{-1}$. The original zonal and meridional velocities simulated in de Marez et al. (2020b) have been here multiplied by two.

The wave ray equation is also modified,

$$\partial_t \mathbf{k} = \partial_\mathbf{x} \omega. \tag{3}$$

$\omega$ is the absolute frequency, $\mathbf{k}$ the wavenumber vector, $\mathbf{u}$ the surface current vector. Bold characters refer to vector notation all along this manuscript.

For this study we consider waves already well developed, far from their generation areas, propagating in the current field without any source term (no dissipation, no nonlinear exchanges between waves, and no wind input, i.e. the right hand side of Eq. (1) is equal to 0). The aim of the current study is to investigate, in a very idealized case, how long waves properties can be modified by an isolated eddy field more realistic than a Gaussian eddy. In a more realistic framework, the waves steepness modified by the current or due to nonlinear waves-waves interactions would lead to local wave breaking as observed in Romero

et al. (2017). Also wind input would generate higher frequency waves which will also interact with the eddy field.

Throughout this manuscript we discuss the evolution of the significant wave height ($H_s$) and the mean wave period weighted on the low frequency part of the wave spectrum ($T_{m0,-1}$), known as "bulk" quantities. We called them "bulk" because they are integrated over the wave energy spectrum, $E(\sigma, \theta)$. They are defined as,

$$H_s = 4\sqrt{\int_{\theta=0}^{2\pi}\int_{\sigma_{min}}^{\sigma_{max}} E(\sigma,\theta)\,d\sigma d\theta,} \tag{4}$$

and

$$T_{m0,-1} = \frac{1}{\int_{\theta=0}^{2\pi}\int_{\sigma_{min}}^{\sigma_{max}} E(\sigma,\theta)d\sigma d\theta}\int_{\theta=0}^{2\pi}\int_{\sigma_{min}}^{\sigma_{max}} \sigma^{-1}E(\sigma,\theta)d\sigma d\theta. \tag{5}$$

The evolution of the wave peak direction ($\theta_p$, $\theta$ where $E(\sigma,\theta)$ is maximum) is also studied. The performance of the wave model used here has already been discussed in boundary currents systems such as in the Gulf Stream, the Drake Passage and the Agulhas current, especially concerning the $H_s$ estimation (Ardhuin et al., 2017; Marechal and Ardhuin, 2021). In those

145 previous studies, wind forcing, waves dissipation, and nonlinear wave-wave interactions have been taken into account.

We initialize simulations with waves that are propagating from the left boundary of a $500$ km $\times$ $500$ km Cartesian domain, with a resolution of $500$ m both in X and Y. The right boundary is open. The initialization is done with narrow-banded wave spectra Gaussian in frequency centered at different peak frequencies, $f_p$=0.1428 Hz, 0.097 Hz, and 0.0602 Hz. The frequencies have been chosen to correspond to the mean periods used in the work of Villas Bôas et al. (2020) (7 s, 10.3 s, and 16.6 s).

The wave spectra have a frequency spread of 0.03 Hz around the peak frequency and an initial $H_s$ equals to 1 m. Waves are generated at the left boundary hourly from the spectra described above. The initial direction of waves is $270°$. The direction convention follows the meteorological convention such that $270°$ waves are coming from the left and are propagating toward the right boundary parallel to the X axis. The wave model global time step is 12 s, the spatial advection time step is 4 s, and the spectral time step is 1 s. The model provides outputs every fifteen minutes. Wave spectra are computed at each grid

point, discretized into 32 frequencies and 48 directions. Fine directional resolution is required for a better description of wave

refraction, especially in strong rotational currents (Ardhuin et al., 2017; Marechal and Ardhuin, 2021). The surface current forcing fields are from de Marez et al. (2020b)'s simulation outputs. In one case we consider the initial shape of the cyclonic eddy (Fig. 1 a,c). In the other case, we consider the fully developed state of the cyclonic eddy (Fig. 1 b,d). The variation timescale of the current is much longer ($\mathcal{O}(1)$ week) than the waves ($\mathcal{O}(1)$ minute), thus the current is assumed to be stationary during one wave train propagation. The simulations forced with the initial eddy is similar to the former works performed over Gaussian eddy (Mathiesen, 1987; Holthuijsen and Tolman, 1991; White and Fornberg, 1998; Gallet and Young, 2014).

The eddy described in the previous section and in de Marez et al. (2020b) is an averaged composite eddy reconstructed from measurements in the Arabian Sea (de Marez et al., 2019). The method of reconstruction tends to an underestimation of the eddy intensity, that is why both the zonal and meridional velocities have been multiplied by two in order to increase the potential effects of currents on wave properties. The eddy is staying geophysicaly realistic (current velocity remains around 1 m.s$^{-1}$ and normalized vorticity lower than 2, see Fig. 1). Those values are comparable with surface vorticity measured in the first hundred metres of Arabian sea (de Marez et al., 2020a) and in other current regimes as in the western boundary currents (Gula et al., 2015a; Tedesco et al., 2019). Although the eddy field represented in Fig. 1b,d is from an averaged composite eddy (solely estimated using in-situ data), it has been considered, in this study, as realistic because it differs from an analytical vortex. Also, the fully developed eddy has been compared with altimeter and drifter data in the Arabian Sea region where it has been estimated. The cyclonic eddy was coherent with those measurements (see Fig. 12, 13, and 14 of de Marez et al. (2019)).

## 3 Results

The frequency sensibility of the incident waves is studied both in the initial and the fully developed eddies. Waves are dispersive in deep water, their group and their energy propagates at the group velocity ($C_g$). For T$_p$=7 s (T$_p$= $\frac{1}{f_p}$), T$_p$=10.3 s and T$_p$=16.6 s, group velocity are 11, 16, and 26 m.s$^{-1}$. To reach X=X$_0$ (a given value of X) short waves take more time than long waves. As waves are generated continuously from the left boundary, a stationary state is reached. The wave field reaches the stationary state after ten hours, nine hours, and eight hours of simulation for initializations of T$_p$=7 s, T$_p$=10.3 s, and T$_p$=16.6 s incident waves, respectively. In Figs. 2, 3, and 4, the fields are taken once the stationary state is reached. Surface currents modulate the wave amplitude, the wave frequency, and the waves direction, the variability of these wave properties are highlighted through the $H_s$, $T_{m0,-1}$, and $\theta_p$ variables. Other aspects of waves' variability, *e.g.,* directional spread or mean direction, are not described here.

### 3.1 Modulation of wave parameters

#### 3.1.1 Significant wave height

Surface currents induce a strong regional $H_s$ variability, especially in a highly solenoidal field (Ardhuin et al., 2017; Villas Bôas et al., 2020; Marechal and Ardhuin, 2021). The presence of the vortex induces strong significant wave height gradients ($|\nabla H_s| = \sqrt{(\partial_x H_s)^2 + (\partial_y H_s)^2}$, noted $\nabla H_s$ hereinafter), inside and outside the eddy (Fig 2). Simulations forced with

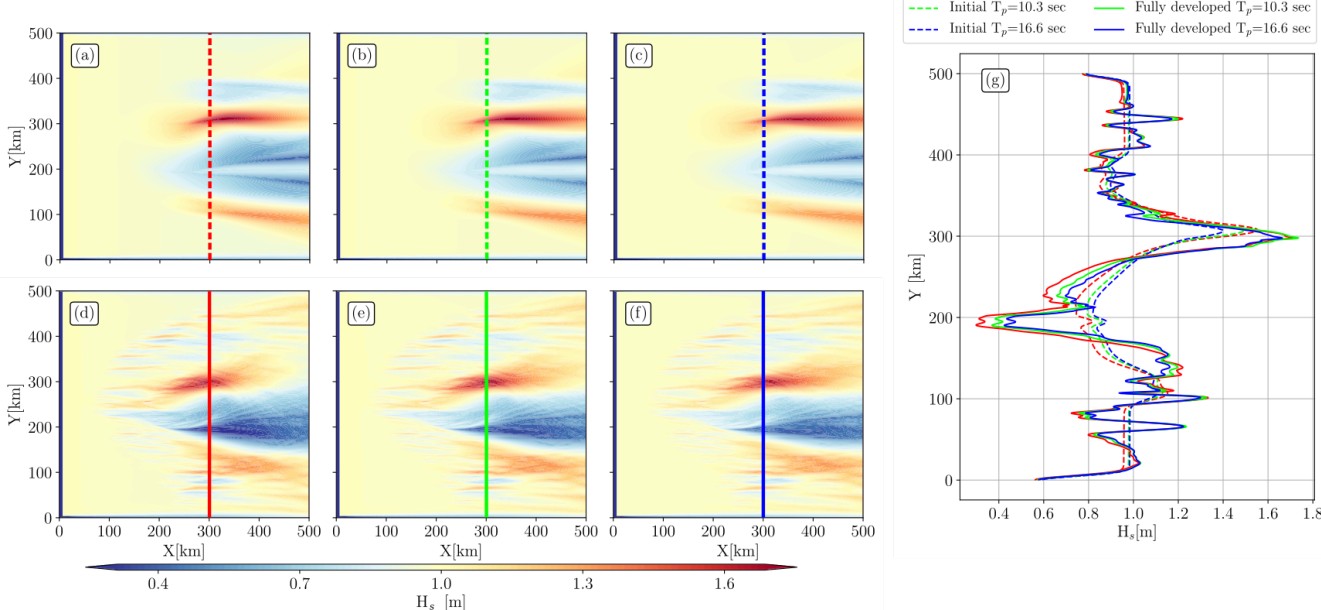

**Figure 2.** Significant wave height ($H_s$) fields for (a, d) $T_p$=7 s, (b, e) 10.3 s, and (c, f) 16.6 s incident waves. Without current forcing the entire domain is equal to the initial $H_s$ (1 m). The first row (a, b, c) shows fields for simulations forced with the initial eddy (Fig. 1(a, c)); the second row (d, e, f) shows the same fields but for simulations forced with the fully developed eddy (Fig1 b, d). Panel g shows $H_s$ along X = 300 km.

the initial eddy (Fig. 2a,b,c) show coherent alternate sign $H_s$ structures along lines of constant X. An important lens shape dipole of $H_s$ increase and decrease is noticeable in the field. $H_s$ reaches a maximum of 1.63 m at X=308 km, 1.62 m at X=324 km, and 1.57 m at X=340 km for simulations initialized with $T_p$=7 s, $T_p$=10.3 s, $T_p$=16.6 s waves respectively. A transect at
X=300 km is given for every initializations in Fig. 2g. Two maximums are visible, the main one at Y=310 km ($H_s \sim$1.6 m) and a secondary at Y=125 km ($H_s \sim$1.2 m). Two minimums are visible, one at Y=200 km ($H_s$= 0.8 m) and a secondary one near Y=380 km ($H_s$=0.85 m). One can see that, at Y=200 km (300 km), shorter incident waves result in lower (higher) $H_s$. Globally, $H_s$ follows the current vorticity signal (Fig. 1c). The enhanced $H_s$ areas are associated to the boundary of the inner eddy core ($\zeta > 0$) and the vorticity ring ($\zeta < 0$) that surrounds the eddy core. Where waves are propagating against the current,
$H_s$ is enhanced which agree with waves-eddies interactions simulated in realistic fields ; (see Fig. 1 of Ardhuin et al. (2017) and Fig. 6 of Romero et al. (2020)).

Simulations forced with the fully developed eddy show a stronger spatial inhomogeneity in the wave field (Fig. 2d, e, f). The initial $H_s$ is more scattered (mostly in the X direction due to the initial direction of the incident wave packet) than in the initial eddy. As noticed for simulations forced with the initial eddy (2a, b, c), the $H_s$ field matches pretty well with the current forcing
(Fig. 1b, d), in other word where surface current gradients are important, strong $\nabla H_s$ are noticed. $H_s$ is mostly modulated by the fully developed eddy core. The modulation of $H_s$ by the fully developed eddy core occurs for X> 50 km which is more

upstream than the $H_s$ modulations induced by the initial eddy. Let us note that $\nabla H_s$ are apparent in the submesoscale eddies that have emerged spontaneously all around the eddy core. In the submesoscale eddy field, the wave field shows alternate sign fluctuations of $H_s$, with globally, the same intensity regardless of the period of incident waves. It is explicitly shown in Fig. 2g at Y<180 km and Y> 350 km for every initialization. In the same transect, at Y=200 km, as previously, for shorter incident waves, the $H_s$ values are lower. However at Y=300 km, the $\nabla H_s$ are almost identical regardless of the periods of the incident waves and the $\nabla H_s$ along Y are strongly sharper for simulations forced with the fully developed eddy with higher extreme values. One can see that $\nabla H_s$ are important dowstream the eddy field. The horizontal size of $H_s$ patches (intensified or decreased $H_s$ structures) are comparable to the width of the eddy (Fig. 2a-f). Finally one can see that for all simulations, the signatures of the eddy in the $H_s$ field are not totally symmetric with respect to the Y axis, whereas the two forcing current field seemed to be so.

The intensity and the patterns of $\nabla H_s$ are very sensitive to the underlying current: the more turbulent the vortex, the sharper the $\nabla H_s$ (Fig. 2). The Fig. 2g shows that, at X=300 km, the (minimum) maximum values of $H_s$ are (lower) higher for the fully developed eddy but are very similar regardless of the periods of the incident waves. For the two currents forcing and all initializations of the model, we computed the $95^{th}$ percentile of the $H_s$ values, the maximum value of $H_s$, and the distance from the left boundary where the maximum value of $H_s$ is located. The results are given in Table.1. Regardless of the periods of the incident waves, the $95^{th}$ of $H_s$ is similar for the two current forcings and varies between 1.18 m and 1.24 m with a maximum of the $95^{th}$ percentile of $H_s$ for simulation initialized with 10.3 s and 16.6 s and forced with the fully developed eddy. The maximum values of $H_s$ are higher for the simulations forced with the fully developed eddy. Finally, the shorter are the incident waves, the closer to the left boundary are the maximum values of $H_s$ with a minimum distance for the simulation forced with the fully developed eddy and initialized with $T_p = 7$ s.

**Table 1.** The $95^{th}$ percentile of the significant wave height ($H_s$), the maximum value of $H_s$, and the distance from the left boundary where the maximum value of $H_s$ is located. $T_p$ is the peak period of the incident waves.

| $T_p$ [s] | 7 | 10.3 | 16.6 | 7 | 10.3 | 16.6 |
|---|---|---|---|---|---|---|
| | Initial eddy | | | Fully developed eddy | | |
| $95^{th}$ percentile $H_s$ [m] | 1.20 | 1.20 | 1.18 | 1.18 | 1.24 | 1.24 |
| Max($H_s$) [m] | 1.63 | 1.62 | 1.57 | 1.73 | 1.74 | 1.68 |
| Distance from the left boundary [km] | 308 | 324 | 340 | 270 | 274 | 280 |

### 3.1.2 Peak direction

The effect of currents on wave direction can be captured to the first order by the $\theta_p$ field. Waves turn in the current field due to the refraction induced by the vorticity of the flow (Kenyon, 1971; Dysthe, 2001). Waves turn toward Y=0 km ($\theta_p$ increase) in the bottom part of the domain and toward Y=500 km ($\theta_p$ decreases) in the upper part (Fig. 3). When waves pass through

the eddy, $\theta_p$ changes due to the vorticity field, at X=125 km for the initial eddy (Fig. 3 a,b,c), and slightly upstream, at X=79 km, for the fully developed eddy (Fig. 3 d,e,f). Patterns showed in Fig. 3 are similar to the $\nabla H_s$ patterns showed in Fig. 2 with a large-scale dipole for simulations forced with the initial eddy and both large and small-scale signal gradients for simulations forced with the fully developed eddy. The peak direction gradient intensity ($|\nabla\theta_p| = \sqrt{(\partial_x\theta_p)^2 + (\partial_y\theta_p)^2}$, noted

$\nabla\theta_p$ hereinafter) depends both on the period of the incident waves and the underlying vorticity field (Dysthe, 2001; Kenyon, 1971). $\nabla\theta_p$ is stronger for simulations initialized with $T_p$=7 s (Fig. 3a,d) than for simulations initialized with $T_p$=10.3 s and 16.6 s with the sharpest gradients for simulation forced with the fully developed eddy (Fig. 3d). In this simulation waves can be deviated by 30° with respect to the initial direction of the waves. The result corroborates Villas Bôas et al. (2020)'s findings where authors forced wave model with synthetic surface currents inverted from Kinetic Energy spectrum (with a random

phase) with different spectral slopes. The more turbulent the current is, the more the waves are refracted. Very long waves trains ($T_p$=16.6 s) hardly reach a deviation of wave direction higher than 10°, both in the fully developed and the initial eddies. Finally one can see that $\theta_p$ differs downstream of the eddy with respect to the initial direction (270°). Downstream of the eddy field, the waves keep in memory the effects of the surface currents passed through.

### 3.1.3 Mean wave period

The surface currents have an effect on the wave frequency (Phillips, 1977). Due to the conservation of the absolute frequency in a current field ($\omega$ in Eq. (2)) , the intrinsic frequency ($\sigma$) is modified which subsequently changes the $T_{m0,-1}$ (Eq. (5)). Wave simulations are initialized with different wave peak frequencies, so $T_{m0,-1}$ is directly impacted. The different initializations of the wave field justify the representation of the relative difference of $T_{m0,-1}$ ($\Delta T_{m0,-1}$) rather than the raw outputs. $\Delta T_{m0,-1}$ is the difference between the outputs of simulations performed with and without surface current forcing. The results are given in

Fig. 4. At first glance, the spatial variability is more striking for simulations forced with the fully developed eddy with patterns similar to the $H_s$ fields (Fig. 2). For the fully developed eddy, $\Delta T_{m0,-1}$ exceeds 3 s in the eddy core for X between 200 km and 400 km. For the initial eddy, for all the initializations, $\Delta T_{m0,-1}$ does not exceed 2 s (Fig 4 a, b, c). Similarly to the $H_s$, the $\Delta T_{m0,-1}$ does not much depend on the period of the incident waves, or at least, not as much as the $\theta_p$ fields studied above. Slight differences are, however, noticeable for simulations forced with the fully developed eddy. It is not clear if there is a link

between the wave period of the incident waves and the slight differences in $\Delta T_{m0,-1}$ shown in Fig. 4g both in the main eddy structure or in the submesoscale eddies. Indeed, $\Delta T_{m0,-1}$ are stronger for long incident waves ($T_p$=16.6 s) in the submesoscale eddies whereas we see the opposite in the core of the fully developed eddy ($T_p$=10.3 s).

  $\Delta T_{m0,-1}$ is positive where waves and current are aligned and negative where waves and current are opposed. This change of $\Delta T_{m0,-1}$ is because the current induces a Doppler shift on the wave frequency (Eq.(2)) and that the absolute frequency

is conserved. If we focus on the maximum of $\Delta T_{m0,-1}$, at Y=200 km, wavelengths increase to about 153 m and $H_s$ values decrease by $\sim 0.65$ m. Where waves and current are opposite we see that $H_s$ are enhanced (Fig. 2) and waves wavelength are shortened and vice versa. It is due to the conservation of wave action ($D_t N = 0$, Eq. (1)). One can see that stripes structures induced by the refraction (Fig. 3) are also visible through the mean wave period fields.

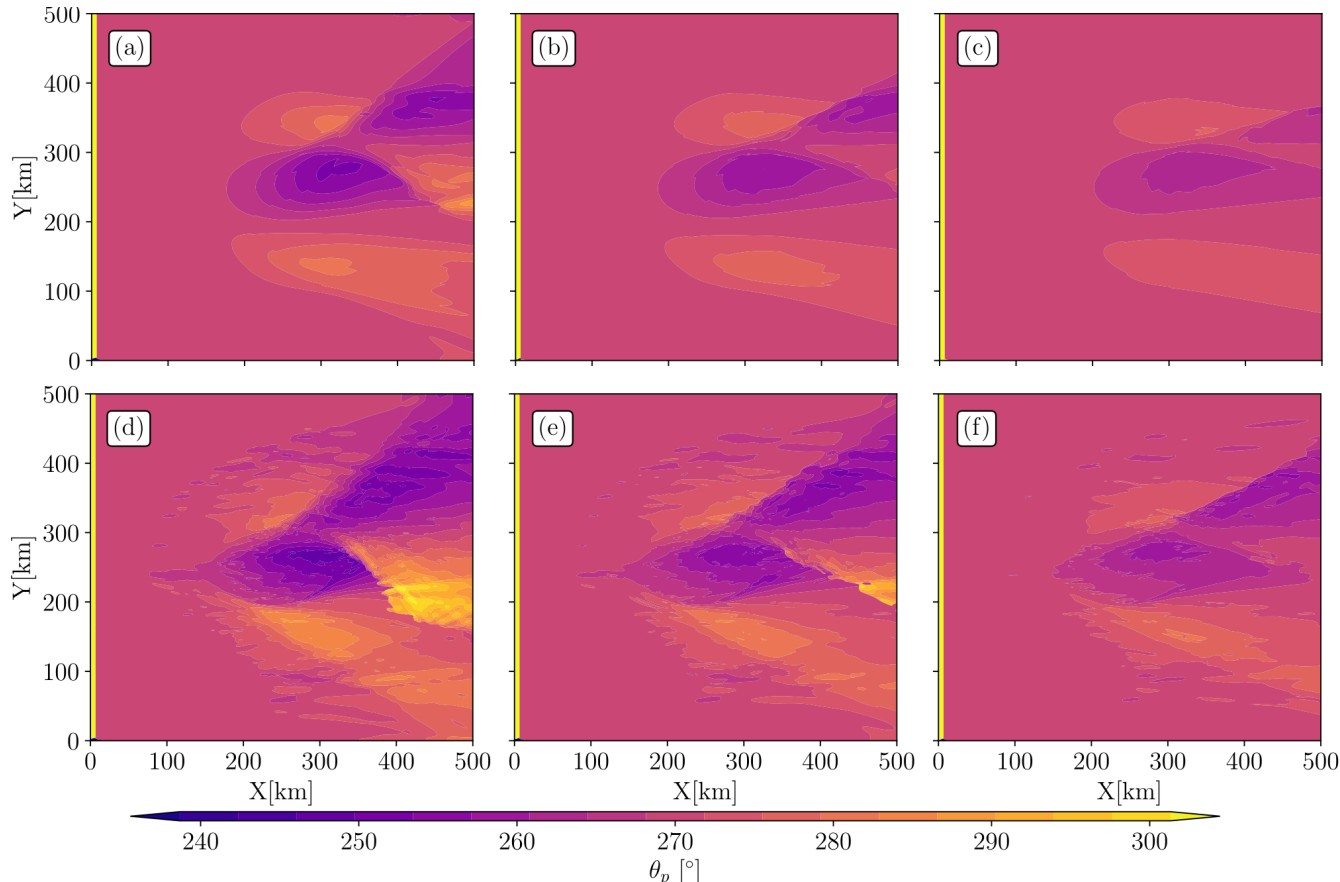

**Figure 3.** Peak direction ($\theta_p$) fields for (a, d) T$_p$=7 s, (b, e) 10.3 s, and (c, f) 16.6 s incident waves. Without current forcing, the entire domain is equal to the initial $\theta_p$ (270°). The first row (a, b, c) shows $\theta_p$ for simulations forced with the initial eddy (Fig. 1 a, c); the second row (d,e,f) shows the same fields but for simulations forced with the fully developed eddy (Fig. 1 b, d.) The narrow yellow bands in the left part of every panels are spurious, they marked the boundary where waves are generated at the left boundary.

We recall that the change of $H_s$ induced by current is due to a superposition of processes. Indeed, in current field, in the

260 absence of wind, the regional $\nabla H_s$ results mainly from the current-induced refraction and the advection of waves action by the current and the group speed (Ardhuin et al., 2017). The current-induced changes in the wave frequency can also increase the $H_s$ (see introduction of Benetazzo et al. (2013)). Note that current refracts waves such that waves and current can becomes aligned (or opposite). So refraction can lead to a change of mean wave period downstream from the refraction areas in the same manner that refraction induce a non-local change of $H_s$.

For all the variable studied here (Fig. 2,3, 4), waves are continuously generated from the left boundary, a solitary incident wave train affect strongly the results presented above, for instance the non-local effect of refraction on the wave field is strongly less pronounced (not shown).

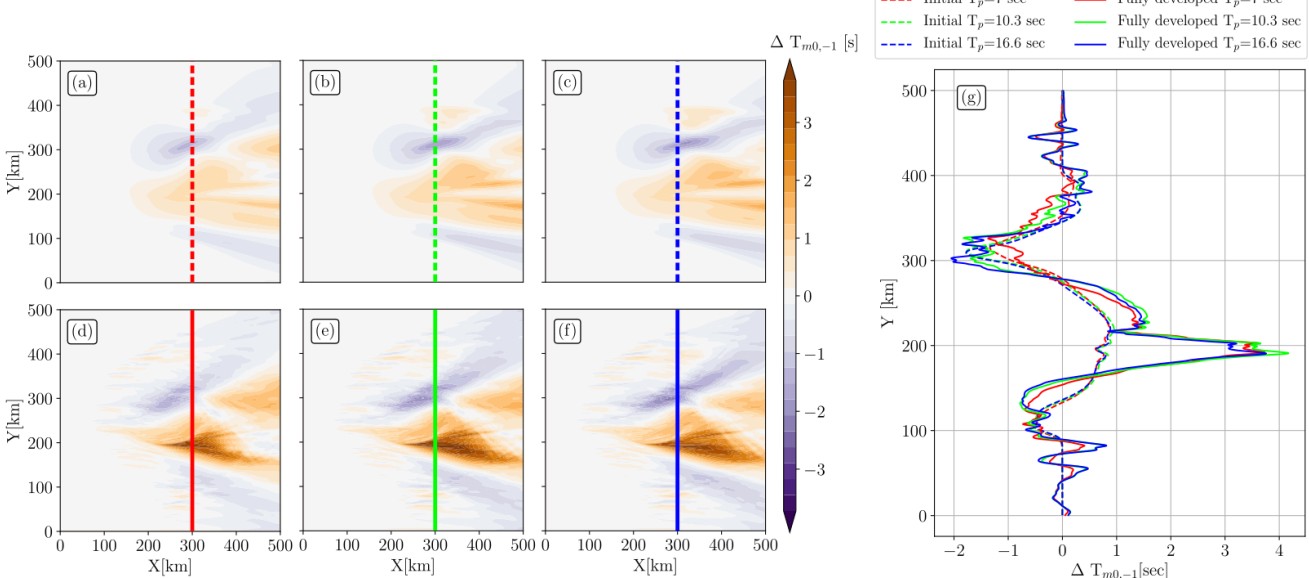

**Figure 4.** Mean wave period difference ($\Delta T_{m0,-1}$) between simulations forced with and without current ($\Delta T_{m0,-1}=T_{m0,-1}$(curr)-$T_{m0,-1}$(Nocurr)). Panels (a, d) show $\Delta T_{m0,-1}$ fields initialized with $T_p$=7 s waves. Panels (b, e) show $\Delta T_{m0,-1}$ fields initialized with $T_p$=10.3 s. Panels (c, f) show $\Delta T_{m0,-1}$ fields initialized with $T_p$=16.6 s. The first row (a, b, c) shows fields for simulations forced with the initial eddy (Fig. 1(a, c)); the second row (d,e,f) shows the same fields but for simulations forced with the fully developed eddy (Fig. 1(b, d)). Panel (g) shows $\Delta T_{m0,-1}$ along X = 300 km.

## 3.2 Ray tracing

Knowing that the wave action ($N(\sigma,\theta)$) is conserved along the wave trajectory in current field (Bretherton and Garrett, 1968),
we show in this section, from a ray-tracing framework, that waves respond very differently to the two eddy fields. In the present study, the isolated vortex refracts the waves and change the wave frequency which leads to a strong inhomogeneity both in the $H_s$ and $T_{m0,-1}$ fields (Fig. 2, 4). The current-induced refraction is highlighted, here, thanks to Monte-Carlo ray tracing simulations, as performed in literature for different current regimes (White and Fornberg, 1998; Ardhuin et al., 2012; Gallet and Young, 2014; Kudryavtsev et al., 2017; Villas Bôas and Young, 2020). This ray tracing method is used in order to follow the conservation of the wave action in the current field. For the ray-tracing, we assume that the surface current is stationary ($\frac{|\mathbf{u}|}{C_g} \ll 1$) and that incident waves are monochromatic. In the real ocean, the wave field is a superposition of wave trains with specific directions and frequencies, thus ray tracing is only a very simplified view of how the direction of the waves are modified by the presence of current.

For the ray-tracing model calculations, the initial direction is 270° (waves are coming from the left boundary) and the initial frequencies are the same than the ones discussed above (T$_p$ =7 s, 10.3 s, and 16.6 s peak periods). We see that the current-induced refraction is sensitive to both the nature of the underlying current and the frequency (or wavelength) of the

incident waves (Fig. 5). The radius of curvature of wave rays is larger where the current field is highly rotational (Fig. 5 d, e, f) and when the ray-tracing simulations are initialized with $T_p$=7 s waves (Fig. 5 a, d). It confirms the works of Kenyon (1971); Dysthe (2001). In the initial eddy, the wave train is refracted both by the eddy's edge (toward the lower part of the domain) and the core of the eddy (toward the upper part of the domain ; Fig. 5 a, b, c). It leads to two wave ray focalisation areas downstream of the initial eddy. These focalisation areas, or caustics, are slightly shifted toward the right boundary when the incident waves are longer. Indeed, the caustic in the upper part of the Fig. 5 (a, b, c) appears at X=330 km, X=370 km, and X=445 km respectively. The locations of caustic formation appear further downstream of the eddy than the location of the maximum values of $H_s$ (Tab.1). However, the position of the caustics are proportional with respect to the location of the maximum values of $H_s$, i.e. the shorter the incident waves, the closer the caustic from the left boundary.

In the fully developed eddy field, both mesoscale and submesoscale eddies refract waves. In comparison with the initial eddy, one can see that the number of caustics increases in the fully developed eddy with a maximum of caustics for $T_p$= 7 s incident waves (Fig. 5d). Even if isolated submesoscale eddies have a vorticity comparable with the eddy core ($\frac{\zeta}{f_0} \sim 1.5$), they do not refract waves as much as the center structure. Indeed, if we look at the southernmost submesoscale eddy we see that one wave-ray deviates about 30 km from the left boundary to the right boundary whereas one wave ray at the center of the domain is deviated of more than 200 km. So, the shape of vorticity patterns is key in the intensity of the refraction. One can note that the ray convergent areas are also localised almost where the maximum values of $H_s$ are spotted (Fig. 2), specially at the edge of the positive vorticity core. The main caustic at Y=300 km is slightly shifted toward the right boundary for longer incident waves which is also qualitatively consistent with results shown in Tab.1.

In a strong rotational current field, the change of $H_s$ is mostly driven by refraction induced by mesoscale and submesoscale currents (Irvine and Tilley, 1988; Ardhuin et al., 2017; Romero et al., 2020). It has been confirmed in additional simulations where the refraction has been deactivated showing maximum values of $H_s$ not exceeding 1.36 m in the core of the eddy field (not shown). With realistic numerical studies in strong current fields, Ardhuin et al. (2012) and Kudryavtsev et al. (2017) showed qualitatively the link between caustics and areas where $H_s$ are enhanced. Here, the ray tracing model highlights the current-induced refraction but it can also explain how the surface currents induces $H_s$ variability. If we assume that one ray is carrying a certain amount of wave action with a certain value of $H_s$ (here 1 m), caustic locations can be assimilated to areas of wave action accumulation and, subsequently, assimilated to areas of increases of $H_s$. If we consider an infinite number of rays, the expected $H_s$ at caustic locations is infinite. However, in a real ocean, because the wave action is distributed in a range of frequencies and directions, these $H_s$ enhancement are limited. In the fully developed eddy, there are more caustics than in the initial eddy due to the submesoscale eddies, it could explain why the $H_s$ fields presented in Fig. 2 d, e, f show more $\nabla H_s$ structures. Also, it partially explain why the extreme values of $H_s$ are very slightly higher for simulations initialized with short waves (Tab.1).

The strong vorticity field, both for the initial and the fully developed cyclonic eddies, induces wave ray scattering which can reach a deviation of several hundred kilometers in comparison with simulations without background current. This deviation is more important for short waves incidence (Fig. 5 a, d). In the ocean, the strong wave-scattering can be responsible for the space-time bias in the forecast of waves' arrival (Gallet and Young, 2014; Smit and Janssen, 2019).

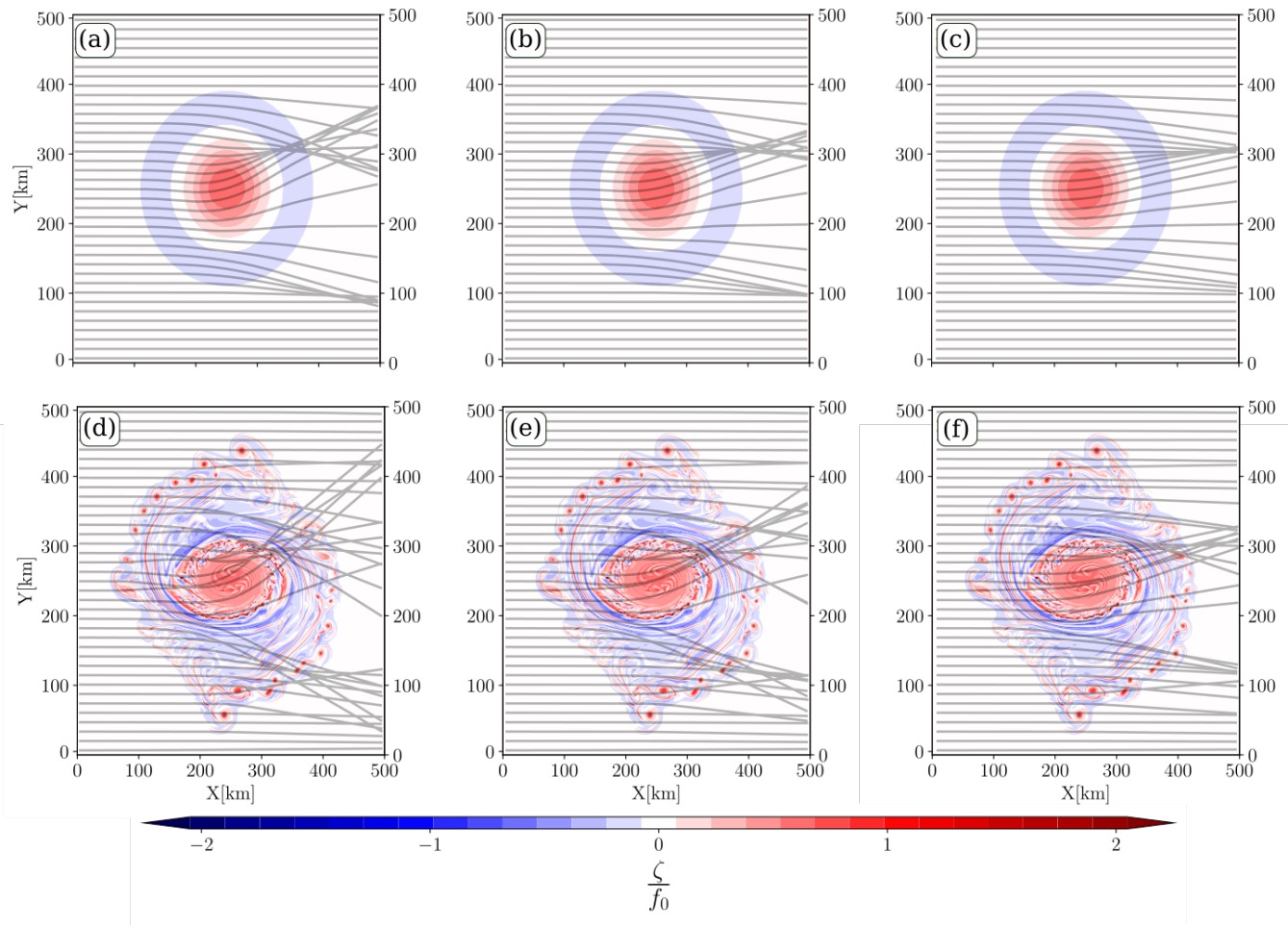

**Figure 5.** (a, b, c) Ray tracing for waves travelling over the initial eddy with $T_p = 7$ s, (a) 10.3 s (b), and 16.6 s (c) peak period. Panels (d, e, f) show the same ray tracing but for waves travelling over the fully developed eddy. The vorticity fields are given in the background.

The present ray tracing simulation shows that refraction have a local effect on wave direction, strong ray deviations appear where surface gradients are strong. However, refraction effects on wave parameters are non-local. Indeed, the sharp $\nabla H_s$ areas seem to be associated to wave ray caustics and can appear both inside and outside the eddies (Fig. 2, 5). In other word, strong $\nabla H_s$ are not necessarily located where strong surface current are spotted.

### 3.3 Is it possible to reconstruct $\nabla U$ *via* the measurement of the $\nabla H_s$?

We have seen that the current-induced refraction and $\nabla H_s$ are driven by the underlying turbulence induced by the presence of the cyclonic eddy. Villas Bôas et al. (2020); Marechal and Ardhuin (2021) showed that at scales between 200 km and $\sim$10 km, $\nabla H_s$ are associated to the nature of the underlying current (structure and intensity). The current intensity gradients

($|\nabla \mathbf{U}| = \sqrt{(\partial_x U)^2 + (\partial_y U)^2}$ with $U = \sqrt{u^2 + v^2}$, noted $\nabla U$ hereinafter), and more specifically the vorticity of the flow ($\zeta$), induces refraction resulting in $\nabla H_s$ patterns correlated to the vorticity patterns (Fig. 1, 2). Note that both $\nabla U$ and $\nabla H_s$ are scalars. Assuming that the group speed of waves is much bigger than the intensity of the current velocity and that the dominant balance in the conservation of wave action (Eq. (1)) is between wave action advection and refraction, Villas Bôas et al. (2020) proposed a scaling between the root mean square (rms) of the vorticity and $\nabla H_s$ (see Eq. (15) of the same

reference). We propose to write the scaling as a function of the wave steepness ($k\langle H_s\rangle$) knowing that $C_g \propto \frac{\sigma}{k}$. It yields the following expression:

$$\frac{\nabla H_{srms}\sigma}{Slope_{KE}\langle H_s\rangle k} \propto \nabla U_{rms}, \tag{6}$$

where $Slope_{KE}$ is the spectral slope of the kinetic energy spectrum (here equal to 3 for the fully developed eddy). The Eq. (6) shows that $\nabla H_s$ is function of surface current gradients, wave steepness ($\langle H_s\rangle k$) and wave incident frequency ($\sigma$). The

motivation of this paragraph is to know if, from high-resolution-wave-height measurements, the nature and the statistics of the flow can be estimated. Today's surface currents measurements from Sea-Level-Anomaly can capture eddy with a shape similar to Fig. 1 a, c (if their lifetime are sufficiently long according to the revisiting-time of altimeters). However, eddies with a more realistic shape (Fig. 1 b, d) are poorly captured (see section 5.2 of de Marez et al. (2020b)). If waves capture information about the current through their interaction with these currents, one can imagine that current signal can be inverted from wave

measurements. It would be relevant for data assimilation in oceanic wave models among other.

Today, filtered altimeter data capture the wave height at fine scale on the global scale (Dodet et al., 2020). The new spectrometer onboard CFOSAT satellite brings a new view of wave measurements from space through directional wave spectra measurements (Hauser et al., 2020). Combining the frequency-direction measurements of CFOSAT and altimeters and knowing the statistics of the surface current at global scale and so the term $Slope_{KE}$ in Eq. (6), the rms of the current gradients

could be estimated. Inverting the wave signal to retrieve surface current properties is not a new concept. To name a few, Rascle et al. (2014) showed that, the images of sea surface roughness from synthetic aperture radars provide clear observations of meso- and submesoscale oceanic features due to the presence of waves. Dugan et al. (2001), thanks to the 3D wave spectrum (wavenumber-frequency), were able to estimate the current speed from the current-induced Doppler shift. Also, Yurovskaya et al. (2019) discussed the possibility to retrieve current from the phase shift spectrum between two successive band measure-

ments provided by the Sentinel-2 satellite. However, all these strategies to infer current gradients are pretty much limited in space.

Thanks to our numerical results, we will test the validity of Eq. (6) in the case of the fully developed eddy. The final aim is to know if the nature of the flow can be estimated by inverting fine resolution $H_s$, $\sigma$ (or k) measurements. For that, we propose to plot the Eq. (6) for the mean state of both wave and current fields, i.e, replace $\nabla H_{srms}$ and $\nabla U_{rms}$ by $\nabla H_s$ and

$\nabla U$ respectively. The mean gradients of the right and the left hand sides of Eq. (6) are shown in Fig. 6. The two fields are plot for the fully developed eddy case and for incident waves fixed at $T_p$=7 s. $\nabla H_s$ and $\nabla U$ have been projected along and perpendicular to the wave peak direction (Fig. 3) respectively.

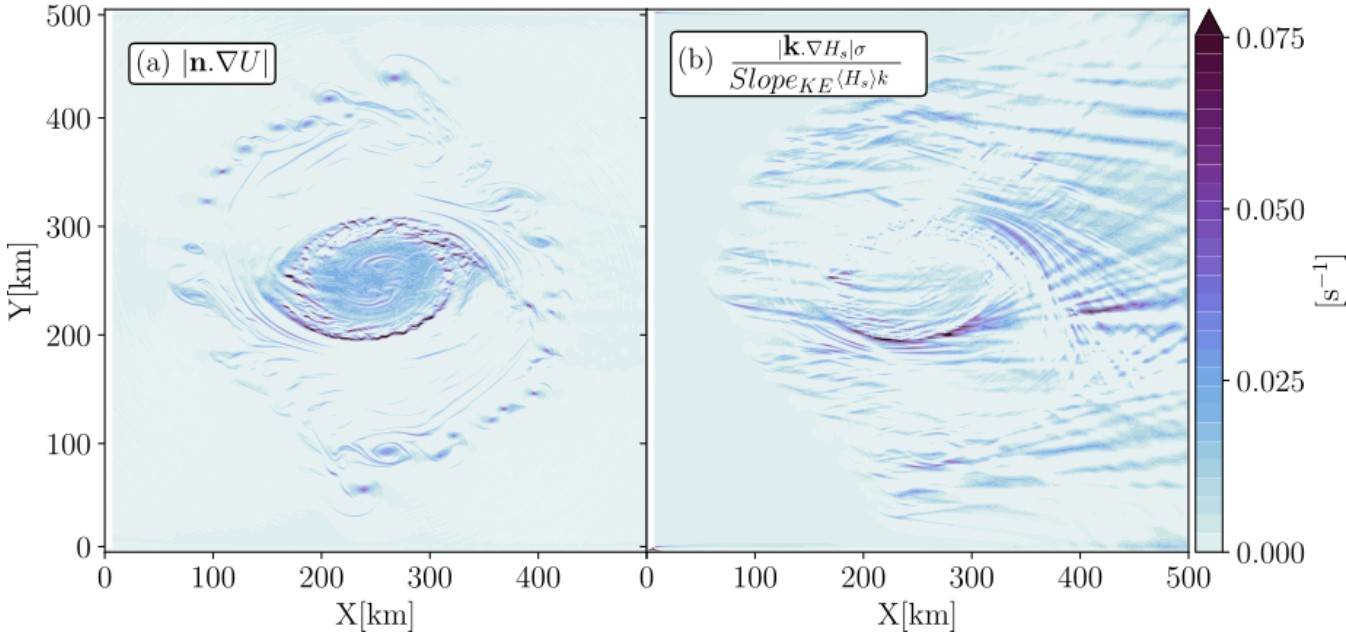

**Figure 6.** (a) Surface current gradients ($\nabla U$) projected perpendicularly to the peak wave direction vector, *i.e.* the right hand side of Eq. (6) and (b) normalized wave height gradients ($\frac{\nabla H_s \sigma}{Slope_{KE} \langle H_s \rangle k}$) projected along the peak wave direction vector, *i.e.* the left hand side of Eq. (6). The two fields are for the fully developed eddy. The panel (b) shows instantaneous field for simulation initialized with $T_p = 7$ s waves.

Both terms of Eq. (6) are of the same order of magnitude (Fig. 6). $\nabla U$ shows rounded structures both for the core of the mesoscale and submesoscale eddies (Fig. 6a), whereas the normalized $\nabla H_s$ field shows more elongated-horizontal structures aligned with the initial wave direction (270°). From X=0 km to X=250 km, the normalized $\nabla H_s$ patterns are aligned with the direction of incident waves; downstream from X=250 km, patterns follow the rays trajectories shown in Fig. 5d. Albeit the two fields show difference of shapes, the two eddy fields are matching both at the mesoscale (the central eddy) and at smaller scales (submesoscale eddies around the core of the ellipsoidal eddy) from X=0 km to X=250 km. $\nabla U$ exhibits fronts at the boundary of the central eddy which is also captured by the normalized $\nabla H_s$ field at Y=200 km. Inside the central ellipsoidal eddy (between Y=200 km and Y=300 km), $\nabla U$ shows a smooth and homogeneous field which is captured in Fig. 6b only between Y=200 km and 250 km. Reader can also see discrepancies in the areas between the central eddy and the submesoscales eddies, where sharp $\nabla H_s$ are shown for Y>300 km, whereas $\nabla U$ are very smooth. Downstream of the eddy, even if $\nabla U$ is null (Fig. 6a), normalized $\nabla H_s$ are very sharp (Fig. 6b).

The normalized $\nabla H_s$ shows similar structures to the surface currents gradient in the first half of the domain, X between 0 km and 250 km (Fig. 6). It is crucial to note that the current gradients estimated from the wave field variability are estimated without any information on the phase of the surface current features. The inversion of the $\nabla H_s$ to infer the underlying surface

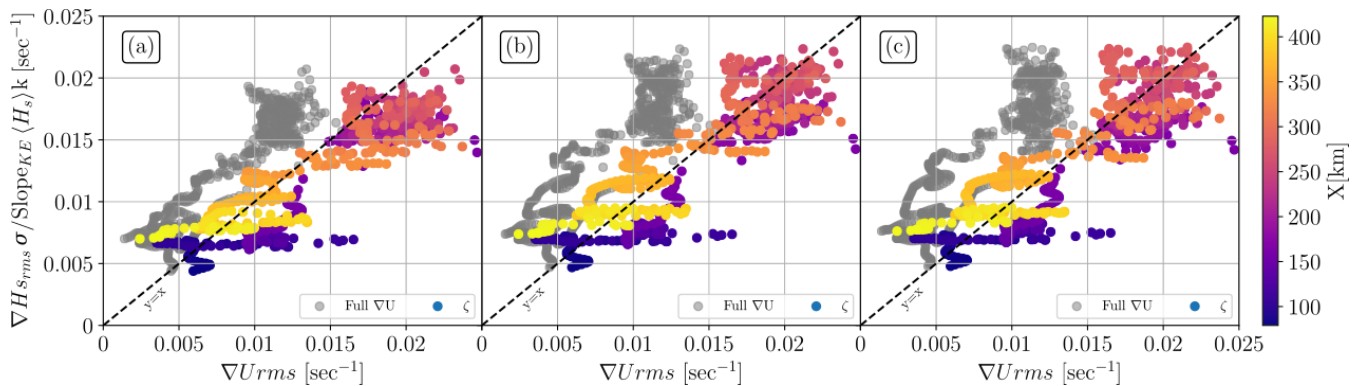

**Figure 7.** Scatter plot of the normalized root-mean-square of significant wave height gradients as a function of the root-mean-square surface current gradients. Colored points are the scatter plot for the vorticity component of the surface current gradients and grey points for the full surface current gradient (diverging component + rotational component). One point correspond to the root-mean-square of the two quantities for a constant X, the value of X is given as colorscale. $\langle H_s \rangle$ is the average value of the significant wave height when simulations reach the stationary state. Panel (a), (b) and (c) are for simulations forced with the fully developed eddy initialized with $T_p$=7 s, $T_p$=10.3 s, and $T_p$=16.6 s respectively.

currents seems to be promising, however both the non-local effect of currents on waves and the initial incidence direction (resulting in a prevalent direction of $\nabla H_s$ patterns) show that the phase of current gradient is hardly reproduced in most of the domain.

In Fig. 7 we illustrate the scaling (Eq. (6)) for all initializations ($T_p$=7 s, 10.3 s, and 16.6 s). The results presented in the figures are for the total current gradient (grey dots) and for the vorticity component of the flow (colored dots). A point in Fig. 7 is the rms of a normalized $\nabla H_s$ and of a $\nabla U$ at fixed distance from the left boundary between X = 79 km and X = 423 km (where current velocity is not null). The normalized $\nabla H_{srms}$ and $\zeta_{rms}$ follow the first bisectrice of the plot unlike the total current gradient ($\nabla U$). For the colored dots, the spread around the first bisectrice is noticeable regardless of the intensity of the

current gradient (or distance from the left boundary) with a maximum of spread at X < 100 km (dark purple dots in Fig. 7). Villas Bôas et al. (2020) proved that $\nabla H_s$ is strongly proportional to the vorticity component of the flow (see their Fig. 12). In the present study, the fully developed eddy is strongly rotational, nevertheless the divergent component of the flow is not negligible ($\delta/f_0 \sim 0.5$, with $\delta$ the relative divergence of the flow). We wanted to show here the effect of the divergence on the proportionality between $\nabla H_s$ and $\nabla U$.

A linear regression is performed between the normalized $\nabla H_{srms}$ and $\nabla U_{rms}$ (and $\zeta_{rms}$) in Fig. 7. For the vorticity field only, the slopes of the regression are equal to 0.72, 0.8 and 0.8 for simulations initialized with $T_p$=7 s, $T_p$=10.3 s, and $T_p$=16.6 s respectively. The R2 score is varying between 0.67 and 0.75 for every initialization. For the full-gradient field (vorticity + divergence of the current), the slopes of the linear regression are equal to 1.13, 1.20, and 1.17 for simulations initialized with $T_p$=7 s, $T_p$=10.3 s, and $T_p$=16.6 s with R2 < 0 for every initialization. This negative R2-score means that the linear regression

fits the data very badly. These R2-scores confirm the results of Villas Bôas et al. (2020) between X=79 km and X=423 km

which states that, in the absence of source terms and for mature incident waves, the variability of the $H_s$ field is mainly driven by the rotational contribution of the current.

Where oceanic eddy destabilizes spontaneously due to horizontal sheared current structures (barotropic instabilities) or vertical buoyancy gradients (baroclinic instabilities, mixed layer instabilities), the resulting ocean surface shows specific $\nabla U$ features. Thanks to wave numerical experiments we were able to observe $\nabla H_s$ structures which are similar to the structures of $\nabla U$ and more particularly to the vorticity component of $\nabla U$. The amplitude of the two gradients are comparable. Knowing the wave incident direction and frequency, it seems promising to invert the waves signal to infer the underlying vorticity field and, perhaps, the instabilities that created such vorticity structures (according to the shape and the size of $\nabla H_s$). Optical instruments have shown the robustness to retrieve the amplitude of the wave field and its associated directional spectrum at fine spatial resolution in a very wide swath (Kudryavtsev et al., 2017). The use of such instrument seems to be a good candidate to capture very small-scale current features by inverting wave characteristics as shown in the fully developed eddy. Also, if the incident wave direction and frequency are known, the same work would be possible with $H_s$ derived from altimeter measurements. Nevertheless there is one drawback, and not least, the non-local effects of current on $H_s$ causing non-zero $\nabla H_s$ where current can be null.

Measuring surface currents from space is a very challenging purpose since past decades (Villas Bôas et al., 2019). Altimetry has proved its robustness to capture surface geostrophic current on global scale by measuring the along track Sea-Level-Anomaly from multiple altimeter missions. The effective resolution of the current products depends principally on the number of satellites. These resolutions have been calculated and show a mean effective resolution coarser than 200 km at mid-latitudes and coarser than 600 km in the equatorial band (Rio et al., 2014; Ballarotta et al., 2019). Even if mesoscale eddies are observable from space (Chelton et al., 2011), surface dynamics at smaller scales are not captured by present altimeter products. As an example, we can cite the small oceanic features in the fully developed eddy (see section 5.2 of de Marez et al. (2020b)). This reality has highlighted the necessity to measure surface currents at finer resolution triggering the emergence of new satellite missions based on innovative measurements methods (Ardhuin et al., 2018; Morrow et al., 2019; Gommenginger et al., 2019; Wineteer et al., 2020). However, even without new current measurements, the wave measurements, which are available both on global scale and at fine resolution, could be assimilated to current models to improve their accuracy specifically for the intensity of the simulated current gradients. Nevertheless, additional works will be required to quantify the non-local effects of currents on $H_s$.

Those current gradients are crucial for a wide range of applications. To cite one example, at front location, where there is a clear contrast in the sea surface temperature field, strong exchanges between the upper ocean and lower atmosphere occur which affect the dynamics of the atmospheric boundary layer (Frenger et al., 2013).

## 3.4 Wave steepness and implications for satellite altimetry

Both $H_s$ and $T_{m0,-1}$ are strongly modulated by the presence of the large cyclonic eddy, which, consequently, modifies the wave steepness ($\mu$). The more turbulent the eddy, the stronger the inhomogeneity in the $H_s$ and $T_{m0,-1}$ fields (Fig. 2, 4). The wave steepness is a key parameter for both recent parametric models of the sea states bias (SSB) in remote sensing measurements

(Badulin, 2014; Badulin et al., 2018) and for the wave dynamics (wave growth, wind drag, wave breaking ; Rapp and Melville (1990); Song and Banner (2002)). Here, we quantify, still for an isolated eddy, the effect of wave-current interactions on the change of the wave steepness at the meso- and the submesoscale ranges. The aims of this section are to highlight the importance of the current effects in the modification of the wave steepness and to discuss qualitatively the implications of those modifications for remote sensing measurements.

$\mu$ is defined in terms of both $H_s$ and wave period (cf. Eq. (3) of Badulin et al. (2018)),

$$\mu = \frac{\pi H_s}{g T^2}, \tag{7}$$

note that the wave steepness ($\mu$) is a dimensionless parameters. We use the mean period ($T_{m0,-1}$) to compute $\mu$ in Eq. (7).

From Eq. (7), we provide the modulations of wave steepness in both the initial and the fully developed eddies when the wave field reaches its stationary state (Fig. 8). The wave steepness is maximum where waves and current are opposed, X $\sim 250$ km,

Y $\sim 300$ km (Fig. 8). The spatial gradients of $\mu$ ($|\nabla \mu| = \sqrt{(\partial_x \mu)^2 + (\partial_y \mu)^2}$, noted $\nabla \mu$ hereinafter) looks more local than the $\nabla H_s$ (Fig. 2a, d). In the fully developed eddy, we can see very localized $\nabla \mu$ at the location of submesoscale eddies. In these areas, the steepness can reach 0.75 which is equal to almost 75% of the maximum steepness spotted in the eddy core. Where waves and current are aligned, the wave steepness is minimum and almost equal to 0 for the fully developed eddy, X $> 250$ km, Y $\sim 200$ km. The maximum values of $\mu$ do not reach very high value ($< 1.2$). In our simulations, the $H_s$ value of incident

waves is equal to 1 m; actually, in the ocean, $H_s$ can be much larger which would multiply $\mu$, presented in Fig. 8, by a factor equal to the $H_s$ of the incident waves. The reader can refer to the Fig. 5 of Badulin et al. (2018) to have an idea of the mean values of $\mu$ measured by the Envisat altimeter on the global scale.

The wave steepness can be estimated on the global scale from altimeter data with different methods, physical (Badulin, 2014) or parametric (Gommenginger et al., 2003). The wave steepness is regionally modified by the presence of the current in

particular for high incident waves (higher than our initialization of 1 m). The fully developed eddy induces stronger $\nabla \mu$ than eddy with a Gaussian shape. The presence of submesoscale eddies leads to the creation of local $\nabla \mu$ (Fig. 8) at scales similar to the submesoscale eddies. As the fully developed eddy is more realistic than the initial eddy, the simulation presented here would help to better understand the quick change of $\mu$ measured by altimeters and better estimate the sea state bias (SSB) in altimeter measurements and provide, perhaps, certain bases for new parametric models of SSB in strong current field. Even

without discussing about the contribution of small scale current gradients, one can see the necessity to take into account current in the estimation of SSB. Indeed, in the present operational SSB models, the wave field is considered as homogeneous at the mesoscale range (Sandwell and Smith, 2005), whereas we see in our simulations that, the wave field is strongly modified due to the wave-current interactions at the mesoscale range, in other words, in both the initial and the fully developed eddy. Finally, because the submesoscale currents induce significant changes in the $\mu$ field, future work on the effects of submesoscale flows

on the SSB is needed.

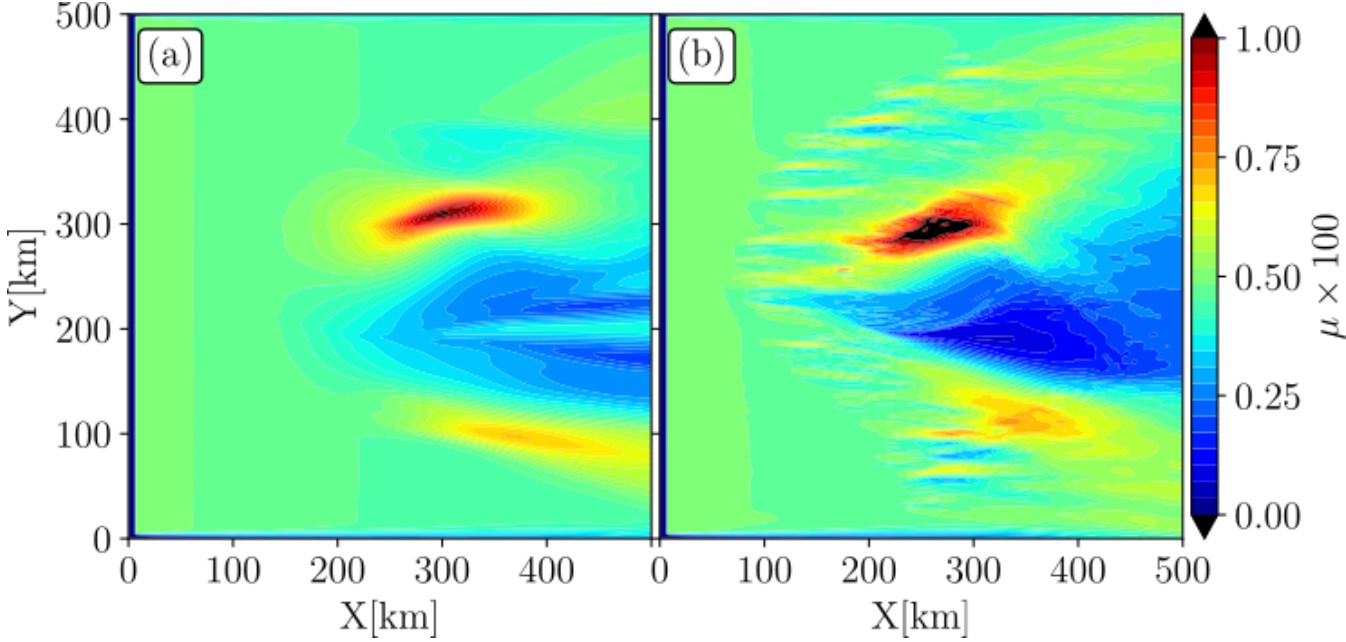

**Figure 8.** Wave steepness multiplied by 100 computed from the mean state of the significant wave height and the mean period in the initial eddy (panel a) and in the fully developed eddy (panel b) for $T_p = 7$ s incident waves.

## 3.5 Effects of broader banded incident spectra and nonlinear wave-wave interactions on wave-current interactions

### 3.5.1 New model setup

In the previous analysis, the incident waves have been simulated with wave spectra Gaussian in frequency with a frequency spread ($\sigma_f$) equal to 0.03 Hz. For time scale much larger than the wave period and assuming that the surface elevation field is a Gaussian process, with negative and positive anomalies around the mean sea level, nonlinear wave-wave interactions lead to a change of the wave energy in the wave field (Hasselmann, 1962). Here we wanted to quantify the effects of nonlinear wave-wave interactions on both $\nabla H_s$ and $\nabla T_{m0,-1}$ in the eddy field. To study the cross-spectral energy flux between frequencies we activate the nonlinear source term ($S_{nl}$). The right hand side of Eq.(1) is thus not equal to 0 any more but to $S_{nl}$. Because simulations initialized with very narrow banded spectrum do not show a clear difference between simulations with and without $S_{nl}$ (not shown), we extend the frequency spread of the initial wave spectra to $\sigma_f$=0.1 Hz.

For sufficiently steep waves, nonlinear wave-wave interactions redistribute wave energy between frequencies over the spectrum which strongly modifies the shape of the spectrum (Komen et al., 1984). As $\nabla H_s$ is function of the wave steepness (k$H_s$, Fig. 6) we expect that nonlinear wave-wave interactions would have an impact on the intensity of $\nabla H_s$. Nonlinear wave-wave interactions are simulated using the discrete interaction approximation (Hasselmann et al., 1985). The wave simulation is run for a sufficiently long time to capture the long term effects of nonlinear wave-wave interactions on the wave parameters. Wave simulation is performed only for 7 s incident waves over the fully developed eddy field. This section is a simple introduction of

how both wave-wave interactions and wave-current interactions could induced inhomogeneity in the wave field, still in a very idealized framework.

More detailed studies will have to be conducted such as with the activation of the other source terms. For instance, activating the wind input source term with a given wind field will have an effect in the high frequency band of the wave spectrum (the development of a wind sea) which, subsequently, will interact with the current field. Also, the presence of both wind and current will modify the wind work at the surface of the ocean. This work is function of the difference between the wind speed and the surface current speed. This relative wind will modulate the wind growth and therefore the wave height in the current field. It would be interesting to scale spatially the effect of this relative wind on the wave parameters ($H_s$, $T_{m0,-1}$). Also, considering the wave dissipation source term will constrain the wave energy in the domain, especially in the areas where the wave steepness are very sharp (Fig. 8).

### 3.5.2 Results

For a given wave parameter ($H_s$ or $T_{m0,-1}$), the relative difference is computed between simulation where nonlinear source term is activated and deactivated (Eq.(8)),

$$\Delta X = \frac{X_{S_{nl}} - X_{noS_{nl}}}{X_{noS_{nl}}} \times 100. \tag{8}$$

The nonlinear wave-wave interactions have a large effect on the spatial gradients of wave parameters studied before, $H_s$ are globally enhanced whereas $T_{m0,-1}$ are decreased (Fig. 9). These changes are more visible where waves and currents are aligned, X>250 km at Y~200 km. The spatial variability of the $H_s$ can reach +80% when $S_{nl}$ is activated. It has been shown that at the same location, wave-current interactions alone showed a strong decrease of $H_s$ (Fig. 2). One can also see that simulation with wave-wave interactions enhances the $H_s$ in the submesoscale eddy field. Globally, we see that taking into account the nonlinear wave-wave interactions, the $H_s$ values increase where the currents decreased the $H_s$ and vice versa. We cannot compare quantitatively Fig. 9a and Fig. 2d (7 s incident waves in the fully developed eddy), because the incident waves have a different spread in frequency.

Nonlinear wave-wave interactions also highlight a change in the $T_{m0,-1}$ field. $\Delta T_{m0,-1}$ shows the opposite spatial variation of $\Delta H_s$. Indeed, where $\Delta H_s$ were (strongly) positive, $\Delta T_{m0,-1}$ are (strongly) negative and vice versa. A transect at X=300 km shows the values of $H_s$ and $T_{m0,-1}$ along the vertical (Fig. 9 c, d). One can see that $\nabla H_s$ are globally reduced due to nonlinear wave-wave interactions especially in the core of the central eddy (Y between 200 km and 350 km). At location of submesoscale eddies, $\nabla H_s$ are also sharper for simulation without $S_{nl}$. $\nabla T_{m0,-1}$ field shows a much more striking difference between simulations with and without nonlinear wave-wave interactions. The transect presented in Fig. 9d shows that $\Delta T_{m0,-1}$ are the most pronounced also in the core of the eddy where $T_{m0,-1}$ increases by 4 s with respect to the mean period at X= 300 km ($\sim$ 8 s) for simulation without source terms. The simulation with $S_{nl}$ shows an increase of $T_{m0,-1}$ values only by 2 s. Whether for $H_s$ or $T_{m0,-1}$, in the current field, wave-wave interactions have the tendency to smooth spatial gradients of the wave parameters driven by wave-current interactions. Here the choice of the parametrization of the nonlinear wave-wave interactions was arbitrary (Hasselmann et al., 1985).

It would be interesting to extend this study to other parametrizations of $S_{nl}$ to better describe how nonlinear wave-wave processes modify regional wave parameter gradients in strong current field. Also, because the wave-wave interactions modify the intensity of the $\nabla H_s$, it would be interesting to characterize again the proportionality between $\nabla H_s$ and the vorticity of the flow (Eq.(6)). In this new numerical framework, considering the nonlinear wave-wave interactions, the R2-score between the rms $\nabla H_s$ and the rms vorticity of the flow drops from 0.67 (simulation without the source term) to 0.42.

This preliminary work on the effects of the source term $S_{nl}$ on the wave field in a realistic eddy field has shown that wave-wave interactions modify the wave field in a current field with strong current gradients. Those nonlinear interactions led to a significant change of wave parameters in the whole domain with the tendency to smooth wave parameter gradients. This work could be extended to other source terms such as $S_{in}$ (describing processes of wave generation due to wind) or $S_{dis}$ (describing a great number of processes of wave dissipation). As an example, we showed in the section 3.4, that the wave steepness ($\mu$) 515 is strongly modified due to the presence of the eddy field. These changes of $\mu$ could lead to an increase of the probability of breaking, subsequently leading to a strong dissipation of the wave energy. It would have large consequences on the potential inversion of the wave signal to estimate the statistic of the underlying current (Eq.6).

## 4    Conclusion and perspectives

In this paper, we studied numerically the effect of an isolated composite cyclonic eddy on the wave properties. Fine resolution 520 wave simulations have been forced with a composite eddy reconstructed from in-situ measurements in the Arabian Sea. The wave model has been forced on the one hand by an initial eddy field with a Gaussian shape and, on the other hand, by a fully developed eddy resulting from the destabilization within the composite eddy. Waves have been simulated by the use of a third generation phase averaged spectral model initialized with narrow wave spectra centered at different frequencies ($T_p =$ 7 s, $T_p =$ 10.3 s, and $T_p =$ 16.6 s). Although wave refraction by an oceanic vortex has already been studied in former papers 525 (Mapp et al., 1985; White and Fornberg, 1998; Gallet and Young, 2014), this study complements studies performed in the past with *(1)* a description of the evolution of the wave bulk parameters (significant wave height and mean wave period) inside and outside the isolated vortex, and *(2)* an investigation of how a fully developed eddy (that really occur in a real ocean) modifies the wave field. Both wave dynamics and kinematics are changed by the presence of underlying currents. These changes are more pronounced where the underlying current has a stronger vorticity. We have shown that the current-induced refraction is 530 stronger for short incident waves and for highly rotational flows. This is consistent with the studies of Kenyon (1971); Dysthe (2001). As the eddies, dynamical at both the meso- and the submesoscale, are certainly more realistic than Gaussian eddies, former studies of interaction between waves and Gaussian eddy underestimate the current-induced refraction, the intensity of $\nabla H_s$ and the wave steepness inside and in the vicinity of an isolated vortex. Those underestimations can have a large impact on the waves forecast but also on the source of noise induced by waves in the ocean level measurements by altimeters (sea-535 states-bias, among other). Thanks to our simulations, we expect that relationships developed in the past for SSB models cannot be applied in strong current gradients. For instance, Tran et al. (2010) proposed to combined altimeter measurements and wave simulations in order to develop a global sea-state bias model. Thanks to the sea-state measurements and period provided

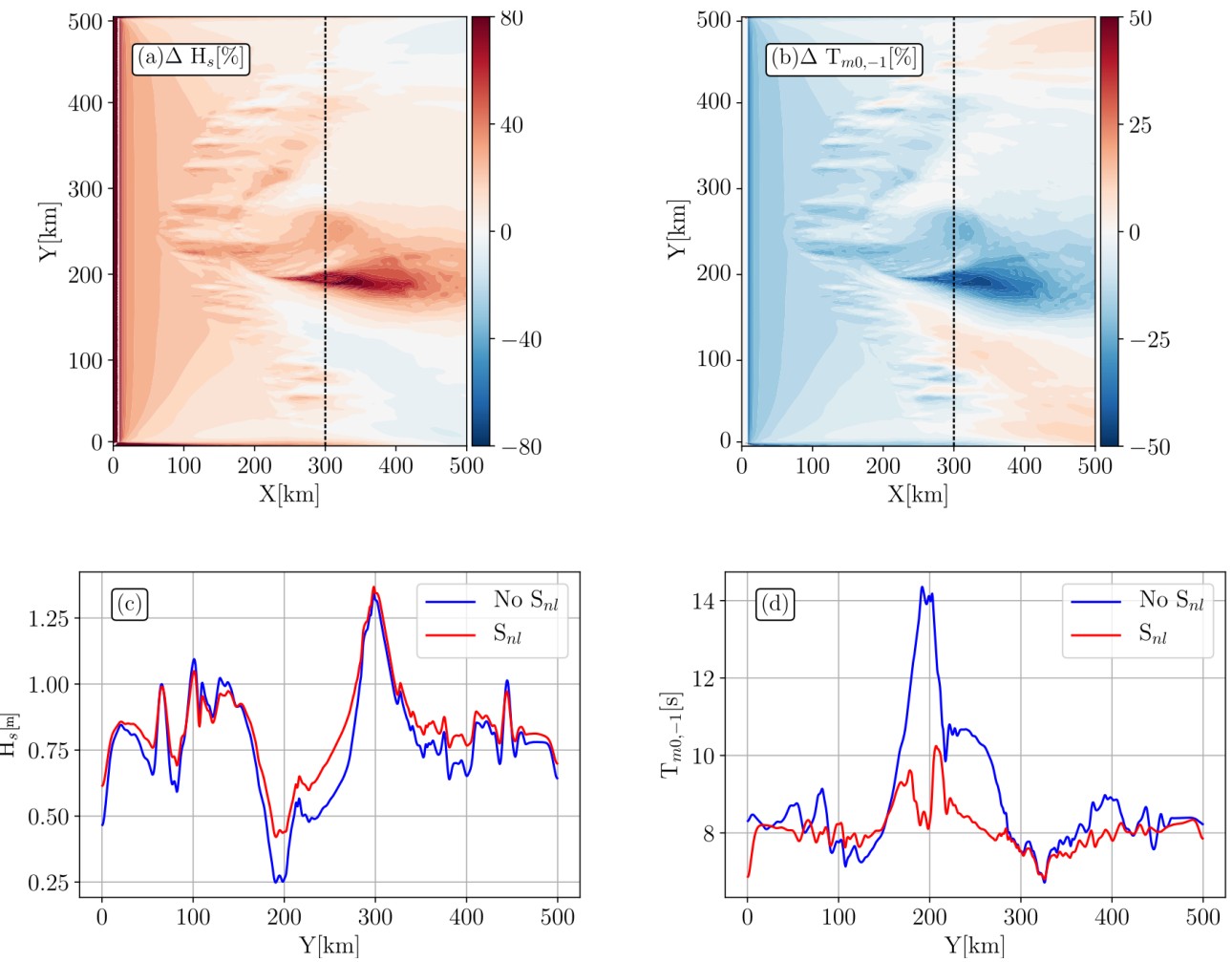

**Figure 9.** Model difference between solutions with and without nonlinear wave-wave interactions. Panel (a) and (b) show the relative difference in percent of the significant wave height and the mean wave period. Panel (c) and (d) show a transect at X=300 km for simulations without (solid blue line) and with (solid red line) nonlinear source term ($S_{nl}$) for $H_s$ and $T_{m0,-1}$ respectively

by wave model (only forced with wind), authors showed the possibility to reduce significantly the error budget in the SSB estimation. However, Tran et al. (2010) parametrized their wave model on a too much coarse grid ($1° \times 1°$) without taken into account current forcing. As we proved here, short-scale currents induce large modifications of wave period at regional scale (smaller than the wind scales). Indeed, in current field, even in a very idealized eddy, $\Delta T_{m0,-1}$ oscillates within 1 s (Fig. 4a-c) and reaches $\sim 3$ s for a more realistic eddy pattern (Fig. 4a-c). So it affects strongly the geometry of the ocean surface through the wave steepness. Redoing the same work of Tran et al. (2010) at finer resolution with current sufficiently resolved would improve their sea-states bias model at regional scale.

Following the relationship introduced in Villas Bôas et al. (2020) based on the balance between wave action advection and current-induced refraction, the significant wave height gradients normalized by the incident wave frequency has been described as a function of the surface current gradients. Besides a good coherence in terms of magnitude between the two quantities, the structures of the normalized significant wave height gradients are very sensitive to the underlying surface current. This work was motivated by the idea to invert wave measurements to infer current properties. We know that measurements of sea level anomaly from space are able to monitor geostrophic surface currents at global scale with a wavelength resolution of several hundreds kilometers in a ice-free areas (Villas Bôas et al., 2019). The total surface dynamics at finer scales cannot be captured by altimeters whereas a lot of oceanic processes occur at those scales (from 1-100 km). This manuscript have shown the possibility inferring the rms of the vorticity of the eddy field from the inhomogeneity in the waves field, as proposed in Villas Bôas et al. (2020). Infer vorticity patterns could allow to capture the small-scale processes (vertical movements, mixing, shear flows...) without measurement of surface currents. Nevertheless, this inversion could not works in the vicinity of a strong $\nabla U$ field because waves keep in memory the previous remote interactions with $\nabla U$ encountered along their propagation. It results in regional inhomogeneities in the wave field, even at the location where current gradients are very weak. The wave inversion is, at the best, only partial. So, the best solution to retrieve the current field at high resolution would be a direct measurement of surface currents from space as proposed in Ardhuin et al. (2018); Gommenginger et al. (2019); Wineteer et al. (2020).

Finally, because the wave-current coupled system is too much complex, much more than the one proposed here, the assumptions proposed in this manuscript are hardly satisfied in nature and the potential effect of the non-linear wave-wave interactions probably as well. An example is, the assumption that submesoscale currents are stationary during wave propagation.

In the present paper, we studied the effect of a turbulent eddy on wave parameters by assuming the underlying current as barotropic in the first metres of the water column. In reality, both the initial and fully developed eddies are strongly sheared along the vertical, particularly in the first five hundred metres (see Fig. 2 of de Marez et al. (2020b)). It is certain that this vertical shear induces a change in the wave dispersion as described by (Kirby and Chen, 1989) and therefore, would modify the wave parameters. Also, because the geometry of surface oceanic features are strongly modified due to the presence of waves (Hypolite et al., 2021), another relevant study would be to study the deformation of the eddy field due to the waves.

*Data availability.* The cyclonic vortex field is available at https://data.mendeley.com/datasets/bwkctkk5bn/1.

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

*Competing interests.* Authors declare no conflict of interest in these works.

*Author contributions.* Model output is available upon request. G.M designed the experiments, performed the numerical simulations and led the analysis of the results and writing. C.d.M provide the surface current fields used as the wave model forcing and contributed to the writing.

*Acknowledgements.* The authors thank B.Chapron for helpful discussions. The authors appreciated the feedback from anonymous reviewers and J.Huthnance (editor) which helped to improve this work. Authors want to thanks their respective funders, G.M is supported both by the Centre National d'Etude Spatiale, focused on SWOT mission and the Region Bretagne through ARED program. C.d.M is funded by Direction Générale de l'Armement (DGA). Simulations were performed using the HPC facilities DATARMOR of "Pôle de Calcul Intensif pour la Mer" at Ifremer, Brest, France. Finally authors thank their respective Ph.D supervisors F. Ardhuin and X. Carton for their thesis guidance all along the past few years.