# Peer review of "Variability of surface gravity wave field over a realistic cyclonic eddy"

_Ocean Science, 2021_

## Author Response (AR1)

We really want to thanks every referees that allow to produce a new version of the manuscript scientifically and grammatically improved. We tried to highlight a better key message. Every elements which were not accurately discussed in the first version are now supported by references, additional figures and examples.

Most of the paper has been rewritten so we advice the reviewers to read again the full text before look at the answers of the first review. Although the answers are given below, some of them (principally minor remarks) are not really answered because of the new structure of the manuscript.

We hope that reviewers will enjoy this new version of the manuscript. We sincerelly thank them for the time that they will dedicate for the second review.

Best regards.

Gwendal MARECHAL (and co-author)

**Asnwers to referees :**

**Colored text are answers.**
* * *
- Title: it is not clear if the study will focus on the scale of the eddy. Reading the manuscript, indeed, it seems that a real distinction between mesoscale and submesoscale is not done, and the eddy is taken as a whole. I suggest, for the title, to focus on the actual subject of the work, that is the variability of the wave field over a realistic cyclonic eddy.

**The title has been modified.**

- Line 2: the wave amplitude is mentioned here, while later (Line 13) the wave height is used as the reference vertical scale. Probably, given the results presented in the study, the use of wave height is more appropriate.

Wave amplitude has been changed to significant wave height.

**The following mistakes have been corrected.**

- Line 17: an "and" between the two references is probably missing
- Line 17: wave height == > significant wave height
- Line 23: gaz == > gas

- I have not a particular suggestion on how to improve the Introduction (section 1), but at the present stage, it seems to be more a list of results instead of a place where briefly introduce the

study in a broad context and highlight why it is important. My suggestion is for a reshaping of this section, to focus on the current state of the research field and key publications.

We have improved the introduction. After the long state of the art of how current affect waves we showed that studying a more realistic shape of eddy will have a significant impact on the Sea-Level Anomaly from space but also on waves forecast.

- Section 2.1 title. I'd rather say: "A realistic cyclonic eddy" (see also my comment below).

In agreement with referee 3 we proposed a new title for this section.

- Line 62: define the variable Bu.

**It has been defined.**

- Lines 69-70: here is mentioned the duration of "half a year", while later (caption of Fig1) the duration is 210 days (more than half a year). Please homogenize and keep throughout the whole paper the actual value used for the simulations.

« Half a year » has been corrected by 210 days.

- Fig. 1. Since the current field (speed and direction) is relevant for the wave model, I'd add two panels showing this variable, which will largely help readers to interpreter the changes in wave parameters.

The current intensity/direction have been added in two new panels in Fig.1

- Fig. 1. what is the meaning of f\_0?

**It has been clarified.**

- Fig. 1. Since the current field is not stationary, is there any reason to choose that specific interval (210 days) after initialization? To me, it seems an arbitrary choice that influences the results and must be carefully motivated in the text. Please add a comment, also about what authors mean for "final state of the simulation" (Line 69).

We gave more informations to explain why we have chosen this current forcing and not another one. It has been clarified in agreement with referee #3

- Figs. 1, 2, and 3. To understand the effect of the eddies on the wave parameters, a comparison with the undisturbed wave field (i.e., no current) is necessary at this stage.

Instead to overload the manuscript, with too much figures, a line has been added in caption of Fig.2 and Fig3 . The difference mean period between simulation with and without current has replaced the instantaneous mean wave period in this new version of the manuscript.

- Page 4. It is defined the surface current, but waves "feel" a wave-averaged current even below the surface. Probably it is not necessary to change the formula, but it is important to specify how waves behave over a realistic current field and that the use of the surface current is an approximation of the real process.

It has been clarified in the wave model setup.

- Fig. 1 and other Figures labelled with X- and Y-axis. Since in the text geographical coordinates are used (i.e., west, north, longitude, ...), I suggest placing them together with labels X and Y on the axes specifications.

« Longitude » and « latitude » have been removed, « west » and « east » as well. We have re-write the manuscript such that paragraphs are consistent with figures axis.

- Page 5. It is not clear how simulations were performed. In particular:

What kind of "narrow band spectrum" was used?

It has been clarified (gaussian in frequency)

**For the three next remarks :**

We have redo all wave simulations such that a new wave train is propagating in the domain every hour. Thanks to this new parametrization waves reach a stationnary state for all initializations (Tp=7, 10.3 or 16.6sec). Details of the new parametrization are given in the paper.

Were simulations initialized with waves travelling from left (west) to right with no boundaries conditions (see the next comment on figures showing the results)?

If so, simulations do not reach a stationary condition, therefore results are representative of a specific time step (as it is mentioned later; indeed, at the given time steps, Hs at X < 100 km is zero, as well the wave period): does this selection affect results and conclusions? Mind also that, because of the different current fields between unperturbed and perturbed simulations, the two wave fields (for a given Tp and time step) do not necessarily correspond to the same "state".

Would have changed the conclusions if, alternatively, the simulation had been done by forcing the field from the boundary and then by trying to reach a steady state for the interior wave field?

- Line 117 "the intensity of the current has been multiplied by five". Does the artificial increase of the speeds cope with the assumption of "realistic cyclonic eddy"?

In term of surface current intensity, multiply by five the initial vortex make the new vortex still realistic. Nevertheless, as noticed by the referee #1 the vorticity do not remains realistic.. We thus multiplied by 2 in stead of 5 the initial vortex of de Marez et al. 2020. The new intensity and vorticity are given in Fig.1.

- Line 122. Cg is the speed of the wave energy.

It has been clarified.

- Line 124. Longitude, without geographical coordinates specified in the Figures, is meaningless (see my comment above).

It has been corrected.

- Line 152: it is not easy from the Figures inspection to appreciate the gradient. A new Figure showing this variable would help. As well it would help a new Figure showing the relative differences between the simulations with different eddies. I let, however, authors decide how to improve the presentation of the results.

The initial Fig 2g has been extended to the mean wave period (Fig.4g). Wave parameters gradients have been described more accurately by adding some indications in the caption of Fig.2, Fig.3 and Fig.4.

- Line 162: what do authors mean by "spurious"? It is shown a value of 360, while it should be 270.

It was due to the fact that only one wave train was propagating in the eddy field. That is why when the unique wave train have propagated over the surface current field the wave direction becomes 0° (=360°). With the new parametrization this large yellow band has been removed because waves are continously emitted from the left boundary.

- Line 265: I wonder how one can obtain the Hs field over such a large area. A comment on the available (or planned) instrumentation would be appreciated. It has been clarified.

Moreover thanks to the new parametrization the conclusion of the paper has changed. A remark on high resolution wave height measurement from space has been added.
* * *
A comparable scaling in terms of the peak wave direction was proposed by Villas-Boas et al. 2021 and used to infer the current gradients. Neither method can accurately invert the current gradients, which is mainly because the effects of refraction are non-local. However, the present work would benefit from applying the scaling of Villas-Boas et al. 2021 to invert the current gradients and comparing them to the current gradients estimated using their method. Also, it would be helpful to apply the current gradient inversions not just for the 7 s waves but to the other cases. Overall, the manuscript needs substantial revisions. Several paragraphs are not well structured. Below I provide several specific comments/suggestions to improve the manuscript.

The relation proposed by Villas Boas et al. 2020 between surface current gradients and significant wave height gradients have been here demonstrated step by step, some lines have been added in the Appendix. The final relation presented in equation 8 is the same than the one proposed by Villas Boas et al. 2020 but in another form. We highlighted that significant wave height gradients are proportional to the wave steepness (kHs), making among other, non linear wave-wave interactions crucial in the intensity of significant wave height gradient. Thanks to referee #3 we have proposed a complementary numerical experiment to show the contribution of the non-linear wave-wave interaction source term in comparison to the results proposed in a very idealised case (S=0). The scatterplot proposed in Villas Boas et al. 2020 (see their Fig.12) have been extended to our study in Fig.7 (for all initial frequencies and with the contribution of the divergence of the flow). The non-local aspect of the current on waves have been even more highligted thanks to the new parametrization of the wave model (waves are emitted from the left boundary continuously every hour rather than studying a unique wave train). The signatures of an isolated eddy on wave parameters reveal that the modulation induced by current on waves have a strong effect downstream the eddy field making the wave parameter inversion to infer current gradient limited.

The following typos have been corrected and expressions reformulated. Some remarks have not been taken into account due to reformulations but most of the proposed semantic have been included. The body of the manuscript has been highly modified so certain expressions have been entirely re-written.

1: small scale --> small-scale. Modified.

8: The word "retrieved" is overly optimistic based on your analysis. You can at most "identify" the current gradients. Retrieve has been removed and replace dby infer/approach/identify

23: gaz--> gas Corrected

26-28: This paragraph is very short. It could be improved. It is unclear what is the paragraph trying to convey. Is it about mesoscales or submesoscales? it has been rewritten.

41: "anticipation" do you mean prediction? It has been changed.

43: "?"? It has been removed.

47: Reads "…numerical model built from Ardhuin et al. (2017) without source terms." What does this mean? Is basically WAVEWATCHIII without source terms. What is the relationship to Ardhuin et al. 2017? Please revise. It has been revised.

56: Parameterization of what? Please explain. We add some lines of the model prametrization where we described how the current field has been obtained.

62: refer --> refers. It has been corrected.

63: Delete "During the simulation,"

92: ". g" --> ", g" It has been corrected.

100: "performances" --> performance. "have" --> has. It has been Corrected

99-102: The studies cited used the source terms (wind input, etc). It is worth specifying. It has been changed.

109: "Indeed, dealing with high" --> High. "allows a better" --> is required for a

110: current --> currents. Sometimes "current" are wiser but most of the time "surface currents" has been used.

114: "the wave one" --> that of the waves. It has not been taken into account. We have reformulate the sentence.

122: There is not need for a new paragraph. I suggest the author revise their use of paragraphs. In several instances the paragraphs are too short. We tried to better define paragraphs according to message in each of them.

122: "are propagating in the current" --> propagate. Also the next sentences ("T\_p= 7 s...than longer wavers") can be replaced with "For T\_p=7, 10.3, and 16.6 s the corresponding group velocities are 11, 16 and 26 m/s.". It has been modfied.

126: "modulate amplitude" --> "modulate the wave amplitude". It has been edited.

127: "respond" --> variability. "waves" --> wave. "for a prescribed underlying current" --> such. It has not been taken into account. We have reformulate the sentence.

Figure 2g – line colors would help distinguish the lines better. Figure has been modified.

132: "Wave train is propagating" --> The waves propagate. It has been reformulated.

148: "occurring" --> apparent. "Occuring" has been changed to "apparent".

150: "actions"--> heights. "is"--> Typos has been corrected.

163: "is function of both" --> depends on It has been corrected.

165: "with perturbed" --> with the perturbed The designation of the two eddy forcing has been modified.

166: insert "the" at the beginning It has been corrected.

167: "the current was turbulent" --> turbulent the current is ->It has not been taken into account. We have reformulate the sentence.

169: "trajectory" --> trajectories. "current" --> currents. It has been corrected.

171: Add "(Kenyon 1971; Dysthe 2001)" at the end. It has been added.

182-184: What does this mean? Please be more explicit in terms of the physical processes..  $\rightarrow$  It has been clarified

185: what do you mean you "guess". Please choose a better scientific word. Also, the text reads "… would be more impacted…" please elaborate more impacted relative to what? We tried to be more scientist in the choose of words.

186: is --> are. Typo corrected

197: Does not make sense. Please revise. It has been corrected.

200: "wave field" --> the wave field. Train --> trains : The typo has been corrected

201: "kinematic: --> kinematics Corrected and modified by wave energy propagation in some part of the paper.

202: "unperturbed" --> the unperturbed Same remark as above.

203: "initial" --> the initial Same remark as above.

205: "Indeed, the" --> the We tried to remove as much as possible link words like "Indeed" but some have been kept.

206-207: delete "confirmed by theoretical works performed by" It has been deleted.

261-=263: please revise is not very clear. This part has been rewritten.

263: "Here, knowing" --> Knowing. Delete "spatial". It has been also rewrittend.

264: "such perturbed" --> an It has been rewritten.

265: delete "that the different". Replace "that occur in" with "within" It has been corrected

266: "approached" you could instead say infer (?). It has been corrected.

263-265: The closing statement is rather promising but is not fully supported by the findings. The analysis does show that you can detect the presence of an eddy. But the details of the structure are not reproduced, which is in part because as mentioned earlier in the text the main mechanism for the modulation of the wave field is non-local.

A long discussion on this point has been given in the discussion section. For example the limit of Eq.8 to infer current gradient from wave field inhomogeneity.

268: Opening sentence of the conclusion is no clear. Please revise. It has been removed

270: frequency --> frequencies Corrected in all the manuscript.

275: delete "all the". Replace "shorts" --> short ''all the" have been removed in the Paper. Short has been corrected.

274-275: You mention the current energy. What about the current gradients. We discussed more about current gradient rather than current intensity.

278: "order" --> terms the use of "order" in such expression has been removed in the paper.

280: delete "until". Replace "dynamic" with dynamics It has been corrected.

281: delete "the used of a constellations of: It has been removed.

283: what do you mean by approach? It has been clarified

Appendix: Equation A4 – what is Cte It has been clarified. It is a integration constant which is removed then.
* * *
**Major comments:**

• Choice of a narrow banded spectrum: What is the effect of the narrow banded spectrum on the wave-wave interaction? Broader banded spectra result from ongoing wave-wave interaction (Hasselmann and Hasselmann 1985, and others). If the spectra are limited to a very narrow band, how does the cross-spectral energy flux change this spectrum over time, even without any perturbation? Does it potentially impact the model results?

A section has been added for this remark at the end of the manuscript. This section is not developed a lot because we wanted to focus only on wave-current interaction in a very idealized framework. We are counting on a second review to know if this section have to be removed or not .

- Sec. 2.1. Even though the model simulations are borrowed from another work, they should be sufficiently described in the method section. Why does it matter the use the full equation of state of seawater? What is the advantage of using this set of equations? The title claims that his simulation is realistic, but it is not explained why. I would suggest that the authors better explain the models' features and advantages.
  - Marez et al. 2019 derive a "composite eddy" this is at least mentioned twice but never explained what that means. If it is a composite of several eddies, how can this be a free model run? How can a composite be realistic and not an average? This needs more explanation.

It has been clarified in the Manuscript, more informations have been added to proved the realistic aspect of the eddy used as model forcing.

• Section 4. I think this is the interesting part of the paper. But it is not well connected to the other parts. I suggest reorganizing the paper such that this analysis is better placed. At the moment, this is neither a result nor really a discussion, it is a deduction on a weak basis:

• Where does eq. (6) appear from? WKB is assumed on what? Would you please state how this relation was derived and where? I see a coherent pattern between both panels of figure 6, but, given this color scale, this is a purely qualitative statement, which similarly appears in other studies.

The demonstration of Eq.8 has been more accurately demonstrated in the Appendix. We also added in the body of the manuscript the different basic elements to obtain this expression.

• I think the simulations allow for a more quantitative assessment of eq. 6. If the gradients of Hs and U "match" (eq. 6), this should be seen in a scatter (regression) between all pixels in Figure 6 a,b. Since a more rigorous analysis in this section is missing, it is hard to follow the rest of the section, which reads like a discussion of possible analysis but not necessarily of this paper (L241 - 267).

We have added new Figure and more comments to better describe the performance of the expression. (eq.8)

• In particular, how can one invert for wave height gradients from observations but not for the surface height directly? Why are the altimeters unable to reconstruct this SSH field? I think the authors miss to say in the beginning that the SSH is a (still dynamics) but average quantity that is not directly observed from a single altimeter track. Altimeters observe the total height changes that appear to be dominated by waves. I think this could also be more clearly stated in the introduction.

As we are not expert of the altimetry we hope that this new version is sufficiently clear.

• I recommend using "initial/ linear" and "fully developed/resolved" currents rather than "unperturbed" and "perturbed" here and throughout the text. Both current fields are perturbations to the incident waves. The linear eddy is somehow a representation of the under-resolved eddy conventional altimeters might see, while the turbulent eddy field is a better approximation of reality. I think this might be a hidden motivation of the authors but is never clearly stated or mentioned except in the discussion. This train of thought should be introduced from the beginning of the paper. Hence the naming of the different experiments is more than just semantic and rather reflects the structural and communicative problems of this paper.

**We have edited the designation of the different current forcing.**

L 167 - 173. What do the authors try to say here? What do you imply? They mention three principles: Random walk, Fermat principles, and \chi/c\_g for deep water waves. None of these principles are directly referenced nor explained. If waves behave like in optics, how is it related to a random walk? Or are they just arguments from Villas Boas and Young restated? This paragraph should be revised and statements justified, as well as grammatical errors corrected. I would suggest starting with the last sentence as a topic sentence.

Indeed, the message was drawn into too much and not explained processed, we removed/edited this part.

• Title: This is to the authors, but I would suggest something like: "Spatial wind-wave variability from (more) realistic meso- and submesoscale eddies"

The title has been edited as adviced by the referree one.

• L 276 This is not true. You do not give a functional relationship between Hs gradients and U gradients. Eq 6 is a proportionality that is not further accessed, or justified. This statement should be revised or removed.

It has been clarified in the Manuscript.

**minor comments:**

All comments has been taken into account, sentences have been corrected, edited and completed.

L 32 "the ubiquity of eddies is no longer proven" what do the authors mean by that? please rephrase.

It has been rephrased.

L 43. "?" something is missing there.

L 74 what does "surface velocity fields" mean exactly. how is surface defined? Both, currents and waves have a complex vertical structure.

Following Referee#1's remark, we indicated what surface current means. What are the real action of three-dimensional current on wave properties at the air-sea interface.

L 94 T\_{m0,-1} why this complicated name? what stands the -1 for?

This is the output of the numerical model, we explained more this variables outputs. We chosed this variables more than  $T_{m0,1}$  or  $T_{m0,2}$  (weighted on higher frequencies of the spectrum) because waves studied in this analysis are long waves (swell).

L 121 I think what the authors mean is that this section analysis the dependence of the wave field on the complexity of the surface currents and the waves peak frequency. And, that longer waves travel faster ( $c_g = g T / 4 pi$ ). I recommend rephrasing the beginning of section 3.

The new parametrization of the wave model shows a stationnary state for all wave initializations thus this point has been removed.

Fig. 2. The lines are hard to distinguish in panel g. I recommend to use color and show the same colored section in the corresponding other panels to guide the reader. It might be also useful to show the approximate center zero-line of the eddies as a single contour in all panels and all figures to show the position of the mesoscale eddy. caption: use "row" rather then "line"

**It has been modfied.**

L 132 i think "initial" should be "incident. The angle convention is confusing. The direction convention is where the waves are propagating TO or FROM? Is this the mathematical or nautical convention?

The convention has been indicated.

L136 enhancement  $\rightarrow$  increase

It has been corrected.

L 142 Y=[150, 300] I am not sure if this appropriate in this journal. normally this should be spelled out.

**Those kind of expression have been removed**

L146 Here and throughout the text. I would rather talk about different simulations than modelS, since this supposed to be the same model.

We replaced model by simulations where it was necessary.

L146f first the authors talk about stronger spatial inhomogeneity for the turbulent simulation but then say its similar to the linear case. please clarify.

It has been clarified.

L150f These sentences are hard to follow. what do the authors try to say?

It has been rewritten.

L 155 suggest: at the first order  $\rightarrow$  to first order

It has been replaced.

L155 are turning ..  $\rightarrow$  refract in the current field and turn southward .. and northward.

We used the appropriate word. « Turning » has been kept for the description of the refraction process.

L160 / Fig 3. Large yellow striped should be removed.

Removed thanks to the new parametrization.

L164 ".. is stronger for simulations with a shorter peak frequency (Fig. 3a,d)". No need to repeat three float point numbers over and over again in the text.

As this manuscript shows a lot of sub panels we prefered to be as much accurate as possible in the references of Figures.

L175 This is a methods sentence, I would recommend rewriting. Again, what is the purpose of this section?

We explained why we studied the mean wave period. In the conclusion a direct application is given.

L183 "super position of processes" Be more explicit, don't let the reader hang. Name these processes, rather than diffuse the attention to 3 other publications and this whole manuscript.

It has been clarified. Processes are given accurately

L185 Why are guesses about a fully divergent field are made here? Even though, from my understanding, the currents are mainly rotational? Or is this just a restatement of the Villas Boas et. al results? please revise.

It has been corrected, it was a pure mistake on my part

L201 Suggest: "wave kinematic"  $\rightarrow$  "wave energy propagation"

It has been corrected.

L217-218 I think what the authors mean is that (local) refraction by currents has non-local effects for the wave energy. Please revise.

It has been revised.

L282 "This manuscript shown .. " other work that did similar work should be cited here.

Citation has been added.

---

## Referee Report (RR1)

**Review of os-2021-53**

**Variability of surface gravity wave field over a realistic cyclonic eddy**

Gwendal Marechal and Charly de Marez

The manuscript has much improved since the previous submission.
major comments:

- Some paragraphs are rather long and it would help the reader to better introduce them.
- section 6 is kind of a paper in it self. I wonder if extending to a broader spectrum and turning on the S_nl terms does counteract the statements in the previous section (ability to reconstruct grad(Hs), and L 311f).
- section 6: Why is only the non-linear term discussed and not the dissipation term. strongly enhanced wave-steepness will have to lead to significant dissipation as well, which would further erode the signal for potential inversions. The authors say they performed these simulations but did not say what their impact was.
- L 34f: I think this statement is not well qualified. The effect of current on wave statistic is still local and hence the potential impact on air-fluxes as well. I suggest revising the statement

A few additional comments to clarify the outcome of. please see below.

- L 203 ... , for shorter ...
- L 269. Why is this a Monte-Carlo simulation? What is a Monte-Carlo tracing simulation? Please revise, cite, or explain.
- eq. 6: If there is an actual derivation of this formula, you may want to put it in the appendix
- L 377: are the slopes in the figure? What Are the numbers in the brackets?
- L 379: I don't understand that sentence
- Section 5: This section is likely worth keeping but I would recommend restructuring it. Please introduce better why this is relevant
- L 455: What happen when these terms are turned on?
- L 492: I think you mean the current has very strong vorticity gradients?
- L 506: where does this come from? This should be also mentioned and cited in result section when it is derived.
- L 518: null -→ small
- L 521: ... And the potential effect of the non-linear wave-wave interaction probably as well

---

## Author Response (AR2)

Dear Editor,

Please find attached the second review of our manuscript, "Variability of surface gravity wave field over a realistic cyclonic eddy". We expect that this new manuscript will be relevant for different applications, as the assimilation of waves measurements in oceanic models to improve the accuracy of fine-scale current gradients, or to improve the estimation/correction of the sea-state bias in radar remote sensing measurements. Thank you for all your recommendations and corrections for this second review of our paper. Your comments helped to improve significantly the quality of this paper. We sincerely acknowledge you for the kindness of your comments.

In this manuscript it is shown, thanks to idealized wave simulations, that the significant wave height, the mean period and the peak direction of surface gravity waves are strongly modified by the presence of an isolated mesoscale eddy and even more when the geometry of the eddy is realistic with coherent features both at the mesoscale and submesoscale ranges. The proportionality between the surface current vorticity and the spatial gradients of the wave parameters are discussed and operated to know if the statistics of the flow can be inferred from the statistic of the wave field. Because the wave field is strongly modified by the current induced by the realistic eddy field, we discuss about the wave steepness and applications for radar altimetry. It is shown that operational sea-state bias parametric models are based on erroneous assumptions. Finally, we show that the exchanges of wave variance in the wave spectrum, due to wave-wave interactions, have the tendency to smooth the wave parameter spatial gradients.

Please find below all the detailed corrections and the main elements that have been added for this new version.

- Abstract: The abstract has been modified in agreement with the new elements.

- Section 3.1.1: the abular 1 has been added.

- Section 3.2: the section has been partially rewritten, a discussion has been added.

- Section 4: the section has been partially rewritten.

- New section: the section 5 about wave steepness has been added.

- Conclusion: The conclusion has been partially rewritten.

- Appendix: The Appendix has been removed. I have edited the core of the paper accordingly.

All my corrections are in red.

Detailed comments.

Generally: "dynamics" for the noun and "dynamic" or "dynamical" (different meanings) for the adjective. Also "upstream" –> "upstream of", "downstream" –> "downstream of" or "downstream from"; "Gaussian" (upper case G).
We sincerely acknowledge the editor for these typo corrections.

Line 9. "approached" –> "estimated"? "Until" is the wrong word but I am not sure what is your intended meaning.
It has been edited.

Line 10. "small" –> "fine"? (Usual adjectives for resolution are "high" and "low" but I prefer "fine" and "coarse").
We have followed your suggestions throughout the manuscript.

Line 23. "narrow" is wrong. Maybe "close", "strong" or "weak"?
« Finally, Villas Boas et al. 2020, under the same assumptions, emphasized the relationship between the vertical vorticity of the flow and the $\nabla Hs$ . »

Lines 41-42. "variabilities" of what? "multi-scale dynamic" is unclear.
Multiscale has been changed by : dynamical at the meso- and the submesoscale range. And most of the sentences with « variability» have been edited.

Lines 43-44. Sentence is too long so "as well as extreme wave height waves" is not clear. Do you mean that wave heights are underestimated?
The sentence has been corrected in agreement with the new diagnostics proposed in this new version.
Line 52. "approached" –> "estimated"?
It has been corrected.

Line 91. "integrated over a certain depth along the first meters of the water column" – "averaged over the top few metres of the water column"
We have followed your suggestion. The sentence has been edited.

Line 99. "wave group" –> "wave group velocity"
It has been corrected.

Line 119. ". . evolution of the significant wave height Hs and . ."

Lines 120, 125. Are all these suffices of T necessary?
Because the mean period can be weighted on higher moments of the wave spectrum, we choose to specify all suffices.

Line 131. "both in horizontal and vertical directions" –> "in both horizontal directions" (there is no vertical in (1)).
It has been edited. Thank you.

Line 137: "top" (use inverted commas).
It has been corrected.

Line 137. Units for 09:15 etc.? I think this sentence should come later.
We have corrected the sentence with another time notation and shift it at the beginning of the section 3.

Lines 165-166. "The response of other waves variability for this underlying current, as the directional spreading or the mean direction, are not described in this manuscript." –> more simply "Other aspects of waves' variability, e.g. directional spread or mean direction, are not described here."?

We took your advice and re-write the sentence.

Line 172. "along meridians (fixed X−axis)" –> "along lines of constant X"? ("axis" is confusing).

We have rewritten this sentence and took your advice for the other sentences.

Lines 177-178. "One can see that more incident waves are short more are the extremes values measured at constant X." Unclear. Do you mean shorter waves have larger extreme values?

Your remark yielded new diagnostics. This part has been edited.

Line 183. ". . the current forcing"

Sorry for the typo. It has been corrected.

Lines 189-190. "more incident waves are short more $\nabla$Hs are sharp." Unclear. Do you mean shorter incident waves have sharper $\nabla$Hs?

It has been edited for the panel g of Hs at X=300 km. « Shorter the incident waves, sharper the $\nabla$Hs »

Line 191. "zonally" –> "in X"?

It has been corrected.

Line 199. South, North –> +Y, -Y?

It has been corrected. Toward the South=> toward Y=0 km and Toward the North=> toward Y=500 km.

Lines 203-204. Sentence "Narrow yellow bands . ." might be better in figure caption.

We moved the sentence to the figure caption.

Lines 215, 228. "absolute frequency". Is this the "intrinsic frequency" in (2)?

It has been clarified «The surface currents have an effect on the wave frequency (Phillips 1977). Due to the conservation of the absolute frequency in surface current, the intrinsic frequency is modified which in turn changes the $T_{m0,-1}$. »

Line 223. Use the symbol for theta.

Sorry for the typo. It has been corrected.

Lines 224-226. I have trouble understanding this sentence. Maybe "This . ." –> "It . ."

It has been clarified. « It is not clear if there is a link between the incident wave frequency and the slight differences in $\Delta T_{m0,-1}$ shown in Fig 4g both in the main eddy structure or the submesoscale eddies. Indeed, $\Delta T_{m0,-1}$ are stronger for long incident waves whereas we see the opposite in the core of the fully developed eddy. »

Line 230. "waves are extended of about 153 m" –> "wavelengths increase to about 153 m"?

It has been edited.

Line 232. Not "precise".

Sorry, it was translated too quick from French. It has been replace by « recall ».

Line 259. Omit "zonally"?
It has been removed.

Lines 264-266. "is deviated of" –> " deviates by"
It has been corrected.

Line 282. Lines 280-282. The notation is confusing. Gradients are vectors not scalars. If you mean the magnitude of the gradient than use |∇U| etc.
You are totally right. Thank you. It has been corrected in Fig.6 as well.

Equations (7) and (A6) to (A7). Better "Cte" –> "constant".
The Appendix was removed. We have preferred to rewrite the scaling of Villas Boas et al. 2020 in term of wave steepness. The section was edited.

Line 343. "resolution . . more than 600 km" is not clear. "finer" or "coarser"?
It has been clarified. « The resolution of global map of surface currents derived from altimetry has been calculated and show a mean effective resolution coarser than 200 km at mid-latitudes and coarser than 600  km in the equatorial band ».

Line 398. ". . parameters (significant wave height and mean wave period) inside . ."
It has been edited.

Line 408. ". . budget by ∼7.5%. . ."?
The sentence has been modified. « Thanks to the sea-state measurements and period provided by wave model (only forced with wind), authors showed the possibility to reduce significantly the error budget in the SSB estimation. »

<End of the correction>

We hope that you will find this manuscript acceptable for Ocean-Science and we look forward to your comments.

With best regards, on behalf of my co-author,

Gwendal MARECHAL

---

## Author Response (AR3)

Dear Editor,

Please find attached the third review of our manuscript, "Variability of surface gravity wave field over a realistic cyclonic eddy". We want to thank the two referee for their remarks which improved significantly this paper.

Please find below all the detailed corrections (in red).

First referee:

Usually "regardless" –> "regardless of", "downstream" –> "downstream of" (if followed by an object). European/French(!) spelling of "metre" please.
It has been corrected throughout the paper.

Line 17. Better ". . bias; using a coarse-resolution eddy field may severely underestimate . ." ?
It has been edited.

Line 20. "at the superficial" –> "in the surface"
It has been corrected.

Line 52. Perhaps ". . more significant for an eddy field with dynamics in the meso- and the submesoscale range . ." ?

Changed by: We demonstrate that wave field characteristics are strongly modified by the presence of the eddy and that those changes are even more significant for an eddy field with dynamics in the meso- and the submesoscale range.

Line 53. "rather" –> "rather than"
It has been corrected.

Line 134. ". . evolution of the significant wave height Hs and . ."
It has been edited.

Line 203. ". . previously, the shorter the incident waves, the lower . ."
It has been corrected.

Lines 214, 215. ". . 95th percentile of . ."
It has been corrected.

Line 336. "Inverse" –> "Inverting"
It has been corrected.

Line 364. "privileged" –> "preferred" or "prevalent"?
It has been corrected.

Line 365. Omit "the part of"?
It has been removed.

Line 392. "which make emerge ∇Hs" is unclear; maybe "causing non-zero ∇Hs"?

It has been corrected in agreement with your proposition.

Line 421. Better ". . 1 m; actually, . ."
It has been edited.

Line 423. Omit "of the values"
It has been removed.

Lines 433-434. Omit "provide" or "put in place"
I removed put in place.

Line 478. Better "1985); it" or even a new sentence.
I did a new sentence.

Line 493. Better ". . flows, consistent with . ." (avoid "which" if not adjacent to object referred to)
I did two sentences and removed the which.

Line 504. "Redo" –> "Re-doing"
It has been corrected.

Line 505. Simpler ". . would improve their . ."?
It has been edited.

Line 509. "inverse" –> "invert"
It has been corrected.

Line 521. "fact" –> "assumption"?
It has been corrected.

Second referee:

 Some paragraphs are rather long and it would help the reader to better introduce them.
• section 6 is kind of a paper in itself. I wonder if extending to a broader spectrum and turning on the S_nl terms does counteract the statements in the previous section (ability to reconstruct grad(Hs), and L 311f).
This section has been added in agreement to one referee remark during the first revision of this paper. I added a sentence in the section (3.5) to answer to your comment.

"Also, because the wave-wave interactions modify the intensity of the grad($H_s$), it would be interesting to characterize again the proportionality between grad($H_s$)  and the vorticity of the flow (Eq.6).  Without doing the scaling the wave action equation while considering the wave-wave interaction source term, as performed inVillas Boas et al. 2020, the relationship between the grad($H_s$) and the vorticity of the flow reveals a drop of the R2-score to 0.42 (not shown, we recall that R2$=$0.67 for simulations without wave-wave interactions)."

Also, for your information, I verified the proportionality (6) in the case of realistic simulations in the Agulhas current where the source terms (Sin, Snl, Sdis) are activated. In those simulations, the proportionality is still verified with a $r^2$-score equal to 0.53.

• section 6: Why is only the non-linear term discussed and not the dissipation term.

strongly enhanced wave-steepness will have to lead to significant dissipation as well, which would further erode the signal for potential inversions. The authors say they performed these simulations but did not say what their impact was.

In my oppinion, verifying the effects of all the source terms would be very interesting to qualify and quantify their individual effects on wave-current interactions in this idealized framework, but it would necessary to redo the numerical framework with idealized (or realistic) wind field and focusing on the breaking parameters of the sea-states. The choice of focusing on the effect on Snl was principally to answer to the previous referee. I added a paragraph in the section (3.5.2) to answer to your comment.

"This preliminary work on the effects of the source term $S_{nl}$ on the wave field in a realistic eddy field has shown that wave-wave interactions modify the wave field in a current field with strong current gradients. Those nonlinear interactions led to a significant change of wave parameters in the whole domain with the tendency to smooth wave parameter gradients. This work could be extended to other source terms such as $S_{in}$ (describing processes of wave generation due to wind) or $S_{dis}$ (describing a great number of processes of wave dissipation). As an example, we showed in the section 3.4, that the wave steepness $(\mu)$ is strongly modified due to the presence of the eddy field. These changes of $\mu$ ccould lead to an increase of the probability of breaking, subsequently leading to a strong dissipation of the wave energy. It would have large consequences on the potential inversion of the wave signal to estimate the statistic of the underlying current $(Eq.\ 6)$."

• L 34f: I think this statement is not well qualified. The effect of current on wave statistic is still local and hence the potential impact on air-fluxes as well. I suggest revising the statement

I revised slightly this paragraph :

"Surface currents seem to increase the deep-water breaking wave probability and the related air–sea fluxes (Romero et al. 2017, Romero et al 2020.) The reader can refer to the instantaneous numerical outputs of Romero et al. 2020 and notify the local effect of the sharp current gradients on the simulated whitecap coverage (see Fig. 5d and Fig. 5i of the same reference). Wave breaking at the air-sea interface is the major source of momentum and heat exchanges between the atmosphere and the ocean (Cavaleri et al. 2012) or gas and sea spray production (Monahan 1986, Bruch et al. 2021). Therefore, surface mesoscale and submesoscale currents have a significant impact on air-sea fluxes (momentum, gas, heat, sea-spray, ...) through their interactions with the wave field.".

A few additional comments to clarify the outcome of. please see below.
• L 203 ... , for shorter ...
It has been edited.

• L 269. Why is this a Monte-Carlo simulation? What is a Monte-Carlo tracing simulation? Please revise, cite, or explain.
I added an additional sentence to describe the method and some reference as example.

• eq. 6: If there is an actual derivation of this formula, you may want to put it in the appendix

The proportionality is partially described in Villas Boas et al. 2020 and in the supplementary material of the same reference. A future publication is in preparation which will demonstrate this equality analyticaly with a analogy with the Synthetic Apperture Radar imagery technique.

• L 377: are the slopes in the figure? What Are the numbers in the brackets?

It has been clarified in the paragraph starting L.377.

• L 379: I don't understand that sentence
It has been clarified in the same paragraph L.377.

• Section 5: This section is likely worth keeping but I would recommend restructuring it.
Please introduce better why this is relevant
Thank you for your interest about this section. The envidence of the relevance of this section has
been improved at the beginning of the section (3.4) and the section has been reorganized.

• L 455: What happen when these terms are turned on?

I edited the end of the paragraph as follows:

 "More detailed studies will have to be conducted such as with the activation of the other source
terms. For instance, activating the wind input source term with a given wind field, will have an
effect in the high frequency band of the wave spectrum (the development of a wind sea) which,
subsequently, will interact with the current field. Also, the presence of both wind and current will
modify the wind work at the surface of the ocean. This work is function of the difference between
the wind speed and the surface current speed. This relative wind will modulate the wind growth and
therefore the wave height in the current field. It would be interesting to scale spatially the effect of
this relative wind on the wave parameters ($H_s$, $T_{m0,-1}$). Also, considering the wave dissipation source
term will constrain the wave energy in the domain, especially in the areas where the wave steepness
are very sharp  (Fig. 8)."

• L 492: I think you mean the current has very strong vorticity gradients?

The vorticity is still a gradient. However I edited this sentence as follows: "These changes are more
pronounced where the underlying current has a stronger vorticity."

• L 506: where does this come from? This should be also mentioned and cited in result
section when it is derived.

I rephrased it as follows: "Following the relationship introduced in Villas Boas et al. 2020 based on
the balance between wave action advection and current-induced refraction, the significant wave
height gradients normalized by the incident wave frequency has been described as a function of the
surface current gradients. "

• L 518: null -→  small
It has been edited.

• L 521: ... And the potential effect of the non-linear wave-wave interaction probably as
well"

I added your proposition.

"I think that the manuscript could be of value to the community. After a third revision, the
manuscript still needs substantial revisions. Several paragraphs are not well structured, and some
are one sentence long. Similarly, the paper organization could be improved by using the typical
format: introduction, methods, results, discussion, and conclusions. Below I provide addtional
comments and suggestions.
I modified the form of the paper. Thank you for this advice.

Abstract: Specify the range of scales investigated. "Fine scale" means different for different readers.
It has been clarified as follows: "The study of these simulations illustrates how waves respond to the numerous kinds of instabilities in the large cyclonic eddy from a few hundred to a few tens of kilometers."

Equation (6):
The slope_{KE} factor of the proposed scaling by Villas Boas et al. 2020 is the reciprocal (1/slope_{KE} ) to that in equation (6) and Figure 7. In other words, slope_{KE} should be in the denominator on the left-hand side of equation (6). I suggest revising the text and the results in Figure 7, accordingly. Also, comment on the outcomes. Specifically, is the 1:1 relationship in Figure 7 lost after correcting the factor?
Indeed, this is an mistake from my self. Thank you for this review. This mistake was not in my computational diagnostics and in my numerical simulations. All the figures have been corrected as well as the associated text.

Figure 7: The y-label is missing the frequency ($\sigma$)."
The figure has been edited.

Gwendal MARECHAL